# Prediction with Action:
# Visual Policy Learning via Joint Denoising Process

**Yanjiang Guo**[12*], **Yucheng Hu**[13*], **Jianke Zhang**[1], **Yen-Jen Wang**[14], **Xiaoyu Chen**[12],
**Chaochao Lu**[3†], **Jianyu Chen**[12†]

[*]Equal Contribution [†]Corresponding Author
[1]IIIS, Tsinghua University [2]Shanghai Qizhi Institute
[3]Shanghai AI Lab [4]University of California, Berkeley
{guoyj22, huyc24}@mails.tsinghua.edu.cn

## Abstract

Diffusion models have demonstrated remarkable capabilities in image generation tasks, including image editing and video creation, representing a good understanding of the physical world. On the other line, diffusion models have also shown promise in robotic control tasks by denoising actions, known as diffusion policy. Although the diffusion generative model and diffusion policy exhibit distinct capabilities—image prediction and robotic action, respectively—they technically follow a similar denoising process. In robotic tasks, the ability to predict future images and generate actions is highly correlated since they share the same underlying dynamics of the physical world. Building on this insight, we introduce **PAD**, a novel visual policy learning framework that unifies image **P**rediction and robot **A**ction within a joint **D**enoising process. Specifically, PAD utilizes Diffusion Transformers (DiT) to seamlessly integrate images and robot states, enabling the simultaneous prediction of future images and robot actions. Additionally, PAD supports co-training on both robotic demonstrations and large-scale video datasets and can be easily extended to other robotic modalities, such as depth images. PAD outperforms previous methods, achieving a significant 26.3% relative improvement on the full Metaworld benchmark, by utilizing a single text-conditioned visual policy within a data-efficient imitation learning setting. Furthermore, PAD demonstrates superior generalization to unseen tasks in real-world robot manipulation settings with 28.0% success rate increase compared to the strongest baseline. Project page at
https://sites.google.com/view/pad-paper.

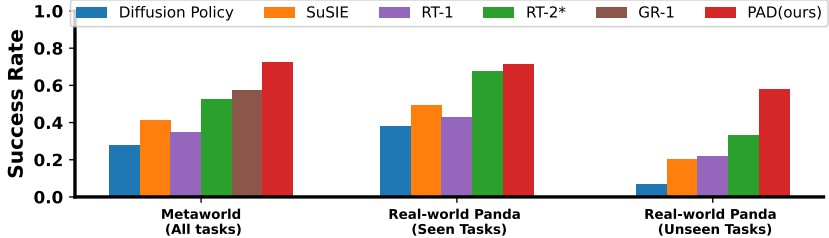

Figure 1: Multi-task performance comparisons in two domains.

## 1 Introduction

Making predictions and taking actions are critical human capabilities, allowing individuals to foresee the change of their surroundings and behave appropriately in response [1, 2]. Despite prediction and action seeming like two distinct abilities, they are highly coupled since they share the same

38th Conference on Neural Information Processing Systems (NeurIPS 2024).

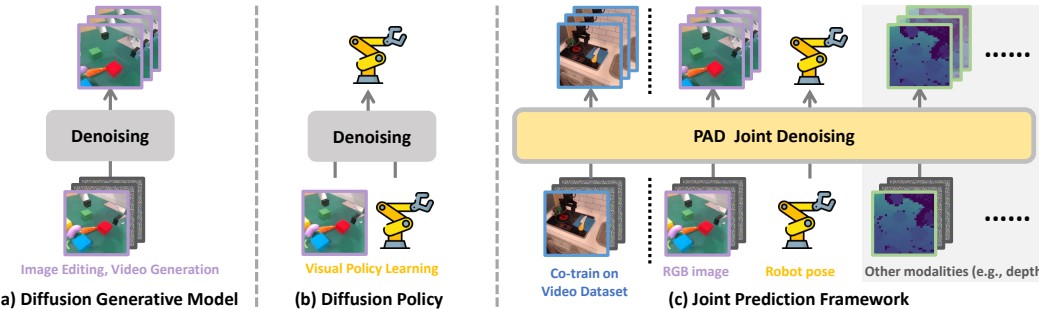

Figure 2: Diffusion models have achieved impressive success in visual generation tasks (a) and visual-motor control tasks (b). Image prediction and robot action are actually highly correlated since they share the same underlying physical dynamics. The PAD framework predicts the future and generates actions in a joint denoising process.

underlying physical laws of the world [3]. Understanding these laws enables humans to make better predictions and actions.

Recently, diffusion models [4, 5, 6] have achieved impressive success in visual generation tasks by training on extensive web-scale image and video datasets [7, 8, 9]. For example, image editing models can predict outcomes based on user instructions [10, 11, 12], while video generation models can generate sequences of future images [13, 14, 15], representing a good understanding of the physical world. On the other line, diffusion models have also shown efficacy in robotic control tasks by denoising actions conditioned on robot observations, known as diffusion policy [16]. Although the diffusion generative model and diffusion policy serve different functions across two domains, we believe that the capability for image prediction could significantly enhance robot policy learning, as they share the same fundamental physical laws. Previous works [17, 18, 19] have employed the image-editing model in an off-the-shelf manner by first synthesizing a goal image and subsequently learning a goal-conditioned policy. However, this two-stage approach separates the prediction and action learning process, neglecting deeper connections between prediction and action. In this way, actions do not leverage the pre-trained representations in the prediction models which encode rich knowledge of the physical world.

In this paper, we introduce the **P**rediction with **A**ction **D**iffuser (**PAD**), a unified policy learning framework that integrates prediction and action under the same diffusion transformer (DiT) architecture [20]. Specifically, we utilize the diffusion transformer model to seamlessly merge all modality inputs and simultaneously predict future images and actions via joint denoising, as illustrated in Figure 2(c). Additionally, the flexible DiT backbone also allows PAD to be co-trained on large-scale video data and extended to other robotic modalities, such as depth images. We have conducted extensive experiments on the MetaWorld Benchmark [21] as well as real-world robot arm manipulation tasks, demonstrating the efficacy of our approach, as shown in Figure 1. Our key contributions are:

- We propose a novel policy learning framework, Prediction with Action Diffuser (PAD), to predict futures and robot actions through a joint denoising process, benefiting policy learning for robotic tasks.

- The proposed PAD framework enables co-training of different datasets containing different modalities, allowing encoding rich physical knowledge from various data sources.

- We outperform previous methods with a clear margin in the Metaworld benchmark, surpassing baselines with a 26.3% relative improvement in success rate using a single visual-language conditioned policy. Furthermore, our method outperforms all baselines in the real-world robot manipulation experiments and can better generalize to unseen tasks.

## 2 Preliminaries

**Problem Statement.** We consider pixel-input language-conditioned robotic control under the imitation learning setting. We denote a robotic dataset $D_{robot} = \{\zeta_1, \zeta_2, ...\zeta_n\}$ comprising $n$ demonstrations. The $i^{th}$ demonstration $\zeta_i = (I_i, l_i, \tau_i)$ contains a natural language instruction $l_i$, a

sequence of pixel inputs $I_i$, and a robot trajectory $\tau_i$ consisted of a sequence of robot poses $p_i^{1:T}$. However, since collecting robotic data is risky and costly, the scale of $D_{robot}$ will be limited. We therefore also consider the RGB video dataset $D_{video}$ which is easily accessible on the Internet. An instance in $D_{video}$ can be represent as $\zeta_j = (I_j)$. Although $D_{video}$ lacks robot action data, our proposed PAD framework enables co-training on both robotic dataset $D_{robot}$ and video dataset $D_{video}$, leveraging the large-scale $D_{video}$ data to enhance visual policy learning.

**Latent Diffusion models.** The core idea of diffusion models is to continuously add Gaussian noise to make a sample a Gaussian and leverage the denoising process for generating data [4]. Let $z_0 = \varepsilon(x_0)$ denote a latent sample encoded from real data. The noising process gradually adds normal Gaussian noise ($\mathcal{N}$) to $z_0$ over $T$ steps, resulting in a set of noisy samples $Z = \{z_t | t \in [1, T]\}$, which is equivalent to sampling from the following distribution: $q(z_t|z_{t-1}) = \mathcal{N}(z_t; \sqrt{\alpha_t}z_{t-1}, (1 - \alpha_t)\mathbb{I})$, where $\{\alpha_t | t \in [1, T]\}$ are predefined hyper-parameters that control the amplitude of the noise. Let $\bar{\alpha}_t = \prod_{i=1}^{t} \alpha_i$, and according to DDPM [5], $z_t$ can be directly obtained by adding a Gaussian noise $\epsilon_t$ to $z_0$: $z_t = \sqrt{\bar{\alpha}_t}z_0 + \sqrt{1 - \bar{\alpha}_t}\epsilon_t$. Further, the denoising process starts with the most noisy latent sample $z_T$, and progressively reduces the noise to recover the real sample $z_0$ with condition $c$. It is based on a variational approximation of the probabilities $q(z_{t-1}|z_t, c)$ given by:

$$p(z_{t-1}|z_t, c) = \mathcal{N}(z_{t-1}; \sqrt{\bar{\alpha}_{t-1}}\mu_\theta(z_t, t, c), (1 - \bar{\alpha}_{t-1})\mathbb{I}),$$
$$\mu_\theta(z_t, t, c) = (z_t - \sqrt{1 - \bar{\alpha}_t}\epsilon_\theta(z_t, t, c))/\sqrt{\bar{\alpha}_t}. \tag{1}$$

The noise estimator $\epsilon_\theta(z_t, t, c)$ is implemented as a neural network and is trained to approximate the gradient of the log-density of the distribution of noisy data [22]., that is:

$$\epsilon_\theta(z_t, t, c) \approx -\sqrt{1 - \bar{\alpha}_t}\nabla_{z_t} \log p(z_t|c). \tag{2}$$

# 3 PAD: Prediction with Action via Joint Denoising Process

## 3.1 Overview of PAD

**Multi-modalities Generation.** In this section, we introduce our PAD framework, which concurrently predicts future frames and actions within a joint latent denoising process. We primarily focus on the RGB image modality $M_I$ and the robot action modality $M_A$. Each robot action can be characterized by a robot pose that includes the position and rotation of the end-effector, as well as the gripper status. Notably, this framework can easily extend to extra modalities $M_E$. For instance, we additionally incorporate the depth image modality in the experiment part, which provides a more accurate measure of distances.

**Conditional Generation.** In the proposed PAD framework, predictions and actions are conditioned on multi-modality current observations, which include RGB images $c_I$, robot pose $c_A$, an additional depth map $c_E$ (in Real-World tasks), and natural language instruction text $l$. The framework simultaneously outputs the corresponding future predictions $x_I, x_E$ and robot action $x_A$. Rather than predicting a single future step, PAD can forecast $k$ future steps $x_I^{1:k}, x_A^{1:k}, x_E^{1:k}$, which can be viewed as $k$ step planning of the robot. Only the first predicted action $x_A^1$ is executed by the robot, which then triggers a new prediction cycle. This iterative prediction and execution process allows the robot to continuously plan and act in a closed-loop manner. The implementation details are discussed further in the subsequent section.

## 3.2 Model Architectures

**Model Input Process.** Given that the original data may come in various formats with high dimensions, we first map all modalities to a latent space and undertake a latent diffusion process. Following the process in [20], the RGB image $x_I$ is initially processed through a pre-trained, frozen VAE[23] encoder $\varepsilon_I$ to derive the latent representation $\varepsilon_I(x_I)$. This latent representation is then converted into a sequence of tokens $t_I$ with embedding size $h$ via tokenizer. Similarly, the robot pose $x_A$ is encoded using a Multi-Layer Perceptron (MLP) [24] into $\varepsilon_A(x_A)$ and linearly transformed into tokens $t_A$ with the same embedding size $h$. If available, the depth image is downsampled and tokenized into $t_E$. The natural language instruction is processed through a frozen CLIP encoder [25] to produce the text embedding $c_l$.

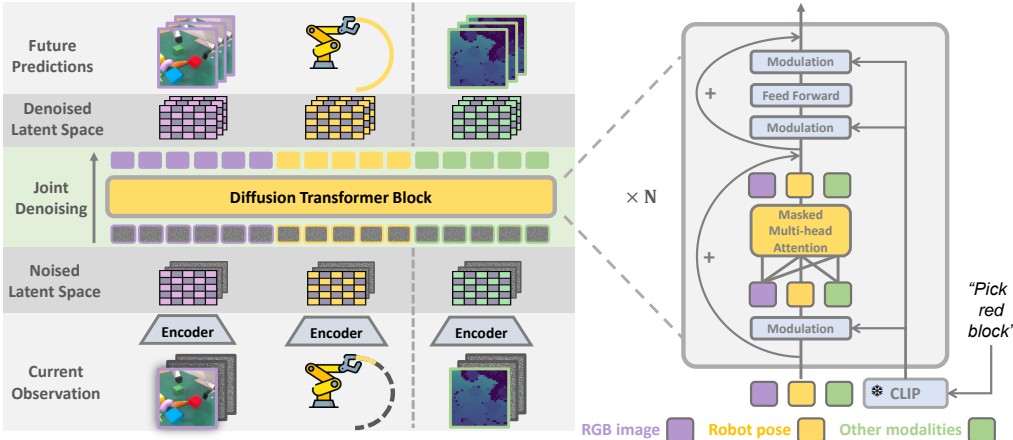

Figure 3: Visualization of the PAD framework. Current observations in different modalities are first encoded into latent and concatenated with white noise channel-wise. These noised latent are then tokenized into tokens and perform a joint denoising process to predict the images and robot actions simultaneously. PAD can flexibly accommodate extra or missing modal inputs through a masked-attention mechanism

**Diffusion Transformer (DiT) Backbone.** We have adopted the Diffusion Transformer (DiT) [20] as our model backbone, which offers several advantages over the U-net backbone commonly used in previous works [18, 17]. Notably, the DiT architecture efficiently integrates various modalities via the self-attention mechanism. Inputs such as RGB images, robot poses, and additional data are transformed into token sequences $t_I, t_A, t_E$ with lengths $T_I, T_A, T_E$, respectively. These token sequences from different modalities are concatenated and undergo a joint latent denoising process

Furthermore, the DiT architecture is adaptable to missing modalities. For example, in the case of a video dataset that lacks robot actions, the input to DiT only comprises the image tokens $t_I$. We simply extend the token sequence to the combined length $T_I + T_A + T_E$ and introduce an attention mask in the self-attention block to exclude the padding tokens. Only effective predictions are retained in the output, discarding any padded parts. A brief illustration of the whole process is depicted on the right side of Figure 3. This design choice enables PAD to be concurrently trained on both RGB-only video datasets and robotic datasets.

**Joint Conditional Generation.** We initialize future observations as white noise and aim to reconstruct future observation frames and desired robot action, conditioning on current observations $c_I, c_A, c_E$. Following a similar strategy as in [26], we concatenate conditional latent and noise latent in the channel dimension. Specifically, after obtaining encoded latent $\varepsilon_I(c_I), \varepsilon_A(c_A), \varepsilon_E(c_E)$, we concatenate these latent with noise to obtain conditioned noised latent $L_I = [\varepsilon_I(c_I), z_t^I], L_A = [\varepsilon_A(c_A), z_t^A], L_E = [\varepsilon_E(c_E), z_t^E]$. For instance, if the encoded latent $\varepsilon_I(c_I)$ has a shape of $c \times d \times d$, then $z_t^I$ would have a shape of $kc \times d \times d$ to represent $k$ future frames, resulting in the final latent $L_I$ having a shape of $(k+1)c \times d \times d$. The other modalities undergo a similar process.

After concatenating the latent, these conditioned noisy latent from different modalities are tokenized into sequences of tokens $t_I, t_A, t_E$ with the same embedding size. The tokenization of image latent $L_I$ follows a patchify process same to [20], while the tokenization of robot pose employs a simple linear projection. Finally, these tokens are fed into multiple layers of DiT to predict the latent representation of future frames. An illustration of the overall process can be found in Figure 3.

### 3.3 Training Process

**Initialization.** Following the initialization process in [15], we also initialize the PAD weights from the DiT model pre-trained on ImageNet for the image generation task conditioned on class [20]. However, we can not directly load the model since we have missing or incompatible model parameters. We discard the label embedding layers in DiT and zero-initialize new layers for text embedding, we replicate the weight of the image latent tokenizer for $k+1$ times to encode the stacked latent, and the encoder and decoder for robot state are also zero-initialized.

**Training Objective.** The diffusion process adds noise to the target encoded latent $\{\varepsilon_I(x_I), \varepsilon_A(x_A), \varepsilon_E(x_E)\}$ and results in noised latent $Z_{I,A,E} = \{z_t^I, z_t^A, z_t^E\}$. We train the PAD model to simultaneously predict the noise $\epsilon^I, \epsilon^A, \epsilon^E$ added to the sample data, conditioned on current observations $C_{I,A,E} = \{c_I, c_A, c_E\}$ and instructions $l$. This denoiser is trained with the DDPM [5] loss:

$$\mathcal{L}_{diff}^{\delta}(\theta) = \mathbb{E}_{\epsilon^{\delta} \sim \mathcal{N}(0,1), t, C, l} \left[ ||\epsilon^{\delta} - \epsilon_{\theta}^{\delta}\left(z_t^{\delta}, t, C, l\right)||_2^2 \right], \tag{3}$$

where $\delta \in \{I, A, E\}$ represents different types of input modalities. The denoising loss $\mathcal{L}_{diff}^{\delta}$ aims to maximize the evidence lower bound (ELBO) [5] while approximating the conditional distribution $p(\varepsilon_{\delta}(x_{\delta})|C, l)$. We jointly minimize the following latent diffusion objectives and use hyperparameters $\lambda_I, \lambda_A, \lambda_E$ to balance the prediction loss between different modalities. Formally, the final training objective is given by:

$$\mathcal{L}(\theta) = \lambda_I \mathcal{L}_{diff}^I + \lambda_A \mathcal{L}_{diff}^A + \lambda_E \mathcal{L}_{diff}^E. \tag{4}$$

## 4 Experiments

In this section, we conduct a series of experiments on the simulated Metaworld Benchmark [21] and a real-world table manipulation suite, utilizing our joint prediction framework. We aim to answer the following questions:

- Can PAD enhance visual policy learning through joint prediction and action with limited robotic data?
- Can PAD benefit from co-training on large-scale internet video datasets and better generalize to unseen tasks?
- Can scaling up computational resources improve PAD's performance?

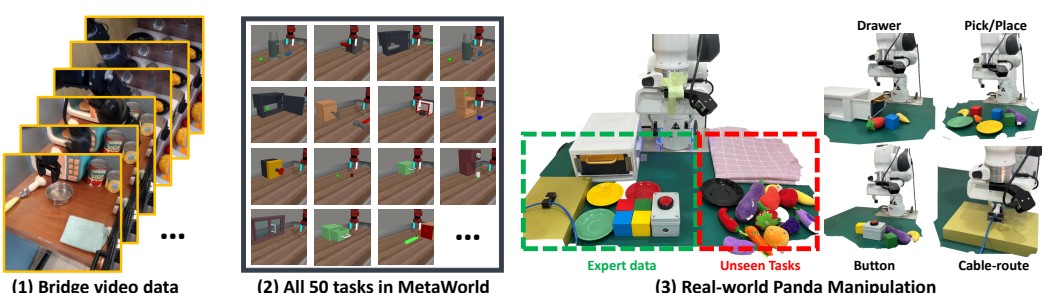

**(1) Bridge video data**  **(2) All 50 tasks in MetaWorld**  **(3) Real-world Panda Manipulation**

Figure 4: We learn a single vision-language conditioned policy to solve all tasks in each domain with limited demonstrations, co-training with the bridge video data. In simulated MetaWorld, we learn a policy to tackle all 50 tasks. In real-world panda manipulations, we split objects into seen objects and unseen new objects to test the generalization ability of our policy.

### 4.1 Environmental Setups and Baselines

**Metaworld.** Metaworld [21] serves as a widely used benchmark for robotic manipulation, accommodating both low-level feature and pixel input modalities. Previous studies that utilized pixel input generally developed separate task-specific policies for each of the 50 tasks. In contrast, our approach demonstrates a significant advancement by employing a single text-conditioned visual policy to address all 50 tasks, within a data-efficient imitation learning framework. We collected 50 trajectories per task, consistently using the "corner2" camera viewpoint and recording the robot's pose with 4-dimensional states that include end-effector position and gripper status. For a fair comparison, we do not utilize an additional depth input in Metaworld.

**Real-World Panda Manipulation Tasks.** Our real-world experiments involve a Panda arm performing diverse manipulation tasks such as pressing buttons, opening drawers, routing cables, and picking and placing with various objects, as shown in Figure 4. We follow the same hardware setup described in SERL [27] and utilize a wrist-mounted camera for pixel input [28]. The robot's poses are represented by 7-dimensional vectors, including 3 end-effector positions, 3 rotation angles, and 1

| Easier Tasks | button-press | button topdown | drawer-open | door-open | faucet-close | plate-slide | reach-wall | window-open | window-close | door-lock |
|---|---|---|---|---|---|---|---|---|---|---|
| Diffusion Policy | 0.92 | 0.16 | 0.36 | 0.32 | 0.76 | 0.60 | 0.72 | 0.60 | 0.36 | 0.12 |
| SuSIE | 0.96 | 0.32 | 0.60 | 0.68 | 0.56 | 0.68 | 0.92 | 0.68 | 0.96 | 0.32 |
| RT-1 | 0.88 | **1.00** | 0.56 | 0.56 | **1.00** | 0.08 | 0.12 | **1.00** | **1.00** | 0.00 |
| RT-2* | **1.00** | 0.84 | 0.92 | 0.96 | 0.96 | **0.88** | 0.76 | **1.00** | 0.96 | 0.40 |
| GR-1 | **1.00** | 0.84 | **1.00** | **1.00** | 0.96 | **0.88** | **1.00** | **1.00** | **1.00** | 0.60 |
| PAD (ours) | **1.00** | 0.92 | **1.00** | **1.00** | 0.92 | 0.72 | **1.00** | 0.92 | **1.00** | **0.88** |
| PAD w/o img | **1.00** | 0.92 | **1.00** | 0.88 | 0.92 | 0.16 | 0.92 | **1.00** | **1.00** | 0.12 |
| PAD w/o co-train | **1.00** | 0.92 | **1.00** | 0.92 | 0.92 | 0.48 | 0.92 | 0.96 | 0.96 | 0.72 |

| Harder Tasks | assem-ble | basket-ball | coffee-pull | hammer | peg-insert | pick-place-wall | shelf-place | stick-push | stick-pull | Average (50tasks) |
|---|---|---|---|---|---|---|---|---|---|---|
| Diffusion Policy | 0.20 | 0.08 | 0.00 | 0.08 | 0.16 | 0.36 | 0.00 | 0.00 | 0.00 | 0.279 |
| SuSIE | 0.40 | 0.24 | 0.32 | 0.04 | 0.24 | 0.24 | 0.08 | 0.16 | 0.16 | 0.410 |
| RT-1 | 0.00 | 0.00 | 0.08 | 0.00 | 0.00 | 0.00 | 0.00 | 0.00 | 0.00 | 0.346 |
| RT-2* | 0.24 | 0.08 | 0.68 | 0.20 | 0.12 | 0.32 | 0.20 | 0.12 | 0.00 | 0.522 |
| GR-1 | 0.64 | 0.08 | 0.52 | 0.48 | 0.24 | 0.48 | 0.28 | 0.60 | 0.44 | 0.574 |
| PAD (ours) | **0.88** | **0.84** | **0.80** | **0.80** | 0.68 | **0.92** | **0.72** | **0.96** | **0.88** | **0.725** |
| PAD w/o img | 0.04 | 0.44 | 0.40 | 0.48 | 0.16 | 0.36 | 0.16 | 0.24 | 0.16 | 0.436 |
| PAD w/o co-train | 0.32 | 0.28 | 0.32 | 0.72 | **0.92** | 0.68 | 0.56 | 0.88 | 0.40 | 0.592 |

Table 1: Comparisons on Metaworld benchmark. We utilize a single policy to solve all 50 tasks in Metaworld. Due to the space limit, we show a subset of tasks and the average success rate on all 50 tasks. Detailed data can be found in Appendix A.4.

gripper status dimension. We collected 200 trajectories per task through teleoperation using a space mouse and scripted commands. Similarly, we developed a single policy capable of addressing all tasks, conditioned on instructions. We also assessed the policy's generalization capabilities on unseen tasks, as depicted in Figure 5.

**Policy Training Details.** As detailed in Section 3, the flexible PAD framework can be co-trained on various internet RGB video data and robotic demonstrations. In order to save computational resources and avoid the need to co-train the model from scratch in each robot domain, we first pre-train the model on internet data to establish better image prediction priors. We then adapt this pre-trained model to various robotic domains, including the simulated Metaworld and the real-world panda manipulation. Empirically, we first pretrain 200k steps on the BridgeData-v2 dataset [9], which consists of 60,000 trajectories. After this, we adapted the model to each domain, continuing training for an additional 100k steps with robotic demonstrations. The pre-training and adaptation stage requires approximately 2 days and 1 day, utilizing 4 NVIDIA A100 GPUs.

Moreover, we found that increasing the weight of the image prediction loss during the early adaptation stages accelerates convergence, as image priors are already established in the pre-trained models. Specifically, we maintained the image prediction loss coefficient $\lambda_I$ at 1.0 throughout the training period and linearly increased $\lambda_A$ and $\lambda_E$ from 0.0 to 2.0 during the 100k training steps.

**Policy Execution Details.** Our policy is conditioned on the current image, $c_I$, and the robot pose, $c_A$, and predicts $k$ frames of futures and actions. We configure the prediction horizon at $k = 3$ and set the interval between frames at $i = 4$ for both Metaworld and real-world tasks. During policy execution, we utilize 75 steps of DDIM sampling [5] to denoise the $k$ steps of future images, $x_I^{1:K}$, and actions, $x_A^{1:k}$. These $k$ step predictions can be viewed as $k$ step planning and only the first predicted action, $x_A^1$, is executed by the robot. The robot then moves to the first desired pose using a simple linear interpolation motion planner, triggering the next prediction cycle.

**Comparisons.** Visual policy learning has been widely explored in previous studies. In our experiments, we opted to compare against a representative subset of prior methods that have either achieved state-of-the-art performance or share a similar architecture with our methods. Notably, all methods are trained on all tasks in the domain **using a single text-condition visual policy**.

- **Diffusion Policy [16].** A novel visual control policy that generates robot actions through an action diffuser. We augmented the original diffusion policy model with instruction conditions to address the multi-task setting. We use the CLIP encoder [25] as instruction encoders, referring to related work [29].

| Task | Button-Press | Cable-Route | Pick (4 tasks) | Place (3 tasks) | Drawer-Open | Drawer-Close | Average |
|---|---|---|---|---|---|---|---|
| **Diffusion Policy** | 0.70 | 0.30 | 0.28 | 0.34 | 0.42 | 0.58 | 0.38 |
| **SuSIE** | 0.74 | 0.44 | 0.46 | 0.40 | 0.42 | 0.70 | 0.49 |
| **RT-1** | 0.72 | 0.52 | 0.40 | 0.34 | 0.44 | 0.50 | 0.43 |
| **RT-2*** | **0.80** | 0.60 | 0.64 | 0.74 | **0.66** | 0.82 | 0.69 |
| **PAD(ours)** | **0.80** | 0.55 | 0.76 | 0.72 | 0.56 | 0.84 | 0.72 |
| **PAD-Depth(ours)** | 0.78 | **0.64** | **0.84** | **0.78** | 0.60 | **0.88** | **0.78** |

Table 2: Comparisons on real-world manipulation in-distribution tasks. PAD achieves the highest success rate. Incorporating depth modality can additionally lead to performance improvement. We evaluate each task with 50 roll-outs.

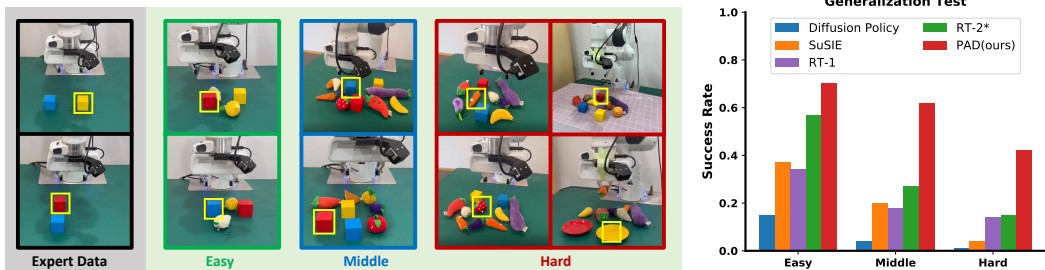

Figure 5: Generalization test under 3 levels of difficulties. The yellow bounding box suggests the target position. Our proposed PAD shows the strongest generalization abilities in unseen tasks.

- **SuSIE [18].** A two-stage approach that utilizes a pre-trained image-editing model [26] to generate image goals for robotic tasks, followed by a goal-conditioned low-level diffusion policy. We fine-tune the image-editing diffusion model on the same dataset and also use the diffusion policy for goal-conditioned behavioral cloning. To ensure a fair comparison, we also use the more powerful DiT framework as the image-editing model.

- **RT-1 [30].** An end-to-end robot control policy that leverages FiLM-conditioned [31] EfficientNet [32] to fuse visual input and language input, then followed by transformer blocks to output action.

- **RT-2* [33] (re-implement).** A large-scale embodied model that directly fine-tunes vision-language models(VLMs) to produce robot actions. The original RT-2 model was fine-tuned on the PaLM model [34], which is not publicly available. Following the specifications outlined in the original paper, we re-implemented the RT-2 model using the InstructBlip-7B [35] backbone.

- **GR-1 [36].** Method that also leverages image prediction to assist policy learning. Different from PAD, they generate images and actions via auto-regressive architecture.

### 4.2 Main Results

**Performance Analysis.** In all comparisons, we train a single visual policy to address all tasks within a domain, conditioned on instructions. Our proposed PAD outperforms all baselines by a significant margin. As shown in Table 1, in the Metaworld benchmark, PAD achieved an average success rate of 72.5%, which markedly surpasses the strongest baseline at 57.4%. Due to space constraints, we present comparisons on a subset of tasks and report the average success rate across all 50 tasks. A comprehensive comparison of all 50 tasks is available in Appendix A.4. Furthermore, Table 2 shows the results in real-world seen-tasks where PAD also attains the highest success rate.

We notice that PAD predicts more precise future images than the GR-1 method (Figure 6), likely due to the superior capabilities of diffusion models in image generation tasks. These precise images may more effectively facilitate policy learning, leading to higher success rates, particularly in tasks requiring precise and accurate operation such as picking small blocks, insertion, basketball, etc., in Metaworld.

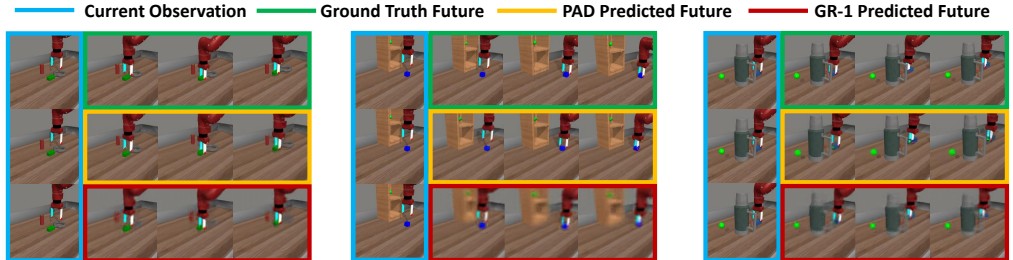

Figure 6: Comparisons on predicted images between PAD and GR-1. PAD generates more precise images than GR-1 which may potentially lead to more accurate control actions. Zoom in for better comparisons.

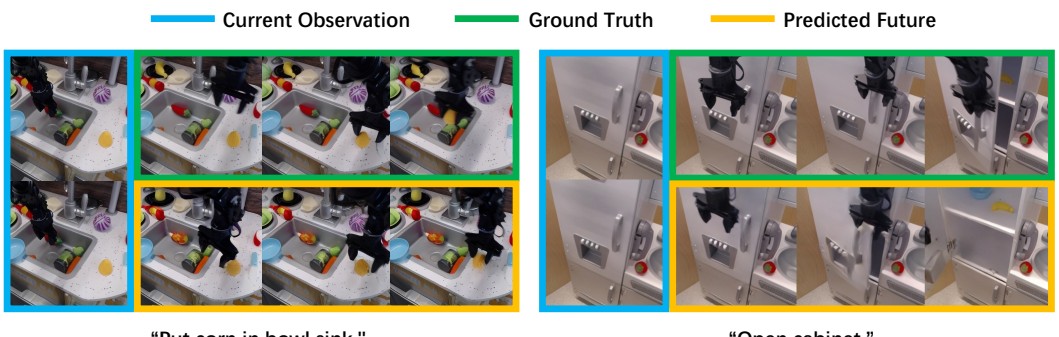

"Put corn in bowl sink."          "Open cabinet."

Figure 7: Predictions on bridge datasets. PAD predicts futures align with instructions but also keeps uncertainty. In the first image, PAD imagines "a yellow pear" instead of the ground truth "banana"; in the second image, PAD imagines scenes faster than the ground truth.

**Quality of the Generated Images.** In addition to achieving the highest success rate in robotic control, we also present visualizations of some image prediction results in Figure 6 and Figure 7. In the Metaworld domain, the predicted image (second row) closely resembles the ground truth image (first row), which is directly decoded from the original latent. In the Bridge domain, the predicted image aligns with the language instructions but also keeps a certain level of uncertainty. These indicate that the PAD model has effectively learned the physical dynamics across these two domains.

### 4.3 Generalization Analysis

PAD can leverage existing physical knowledge from co-training on large-scale internet video datasets to enhance its generalization capabilities across new tasks. We evaluated PAD's generalization ability in real-world panda manipulation with unseen tasks. As depicted in Figure 4, the expert dataset comprises only colored square blocks and plates, while we introduce a variety of previously unseen fruit and vegetable toys during testing. We designed tasks of three difficulty levels: easy mode, featuring 1-4 disturbance objects; a middle level with 5-15 disturbance objects; and difficult tasks that require picking previously unseen objects with 5-15 disturbances or unseen backgrounds. We excluded depth input to ensure a fair comparison. As illustrated in Figure 5, PAD demonstrates remarkable generalization abilities, successfully managing out-of-distribution objects such as strawberries, carrots, and eggplants, and even adapting to new backgrounds. The baseline method failed to generalize to difficult unseen tasks.

### 4.4 Ablation Studies

**Effectiveness of RGB image prediction.** We evaluated the effectiveness of our joint prediction process by modifying the original model to **exclude** the image prediction component, namely in PAD w/o image prediction. This modification leads to significant performance drops compared to PAD, as illustrated in Table 1. The absence of image prediction compromises the robot's ability to utilize the physical knowledge encoded in the image modalities, which may be crucial for robotic control.

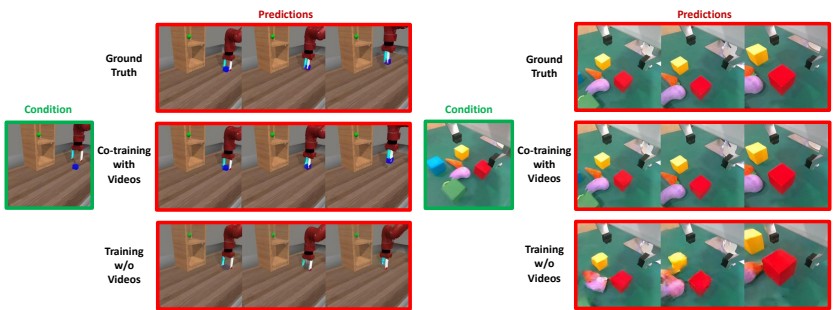

Figure 8: We observe that co-training with an internet video dataset leads to better image generation qualities, which may potentially lead to better robot action predictions.

Furthermore, predicting solely the robot pose provides only low-dimensional supervision signals, potentially leading to overfitting of the training data.

**Effectiveness of Co-training with Internet RGB Video Datasets.** Another major benefit of PAD is the ability to co-train with large-scale internet-sourced RGB videos, which potentially leverages the physical knowledge embedded in these web datasets to enhance robotic pose prediction. We train PAD without the inclusion of web-scale RGB data, namely PAD w/o co-train. We observed a performance drop without co-train on the video dataset, as shown in Table 1. Furthermore, the quality of the predicted image also decreased. For instance, as depicted in the bottom column of Figure 8, the blue block is absent in the predicted images. The quality of the predicted images markedly improves with co-training, which in turn indirectly enhances robot action prediction.

**Compatible with Additional Modalities.** As detailed in Section 3, our framework accommodates additional modalities owing to the adaptable DiT architectures. We incorporate additional depth image inputs in real-world manipulation experiments and jointly predict future RGB images, depth images, and robot actions, denoted as PAD-depth. We observe highly aligned prediction results among different modalities under our joint denoising framework, with some results illustrated in Figure 9. The inclusion of depth input enhances performance in manipulation tasks, as demonstrated in Table 2. This improvement may stem from the precise prediction of depth information, which aids agents in discerning distance changes, thereby enhancing performance. Moreover, our framework could be extended to predict other modalities relevant to robot control, such as tactile force or point clouds, which we left for the future work.

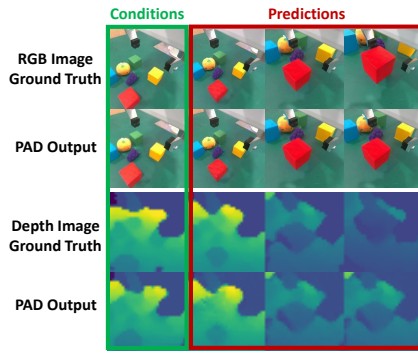

Figure 9: PAD can flexibly train with additional modality, and simultaneously predict all the futures through joint denoising process.

### 4.5 Scaling Analysis

We evaluated models across various sizes and patchify sizes [20], as outlined in Table 3. For example, the $XL/2$ model denotes the model with an $XL$ size and a $2 \times 2$ patchify size. Halving the image patch size will quadruple the image token lengths, which leads to higher computational costs. Our findings reveal a strong correlation between computational allocation (measured as transformer Gflops) and the success rate (SR) of the learned policy, as depicted in Figure 10. All the experiments are run in Metaworld benchmarks and detailed success rates for each task are provided in Appendix A.5.

## 5    Related Work

**Pre-training for Embodied Control.** Vision-language pre-trained models, encoded with physical knowledge, can enhance embodied control from multiple aspects. Primarily, the pre-trained model can directly act as policy by either generating high-level plans [37, 38, 39, 40, 41] or producing direct

|  | PAD-XL/2 | PAD-XL/4 | PAD-XL/8 | PAD-L/2 | PAD-B/2 |
|---|---|---|---|---|---|
| **Layers** | 28 | 28 | 28 | 24 | 12 |
| **Hidden size** | 1152 | 1152 | 1152 | 1024 | 768 |
| **Heads** | 16 | 16 | 16 | 16 | 12 |
| **Token length** | 257 | 65 | 17 | 257 | 257 |
| **Parameters** | 661M | 661M | 661M | 449M | 128M |
| **Gflops** | 119.1 | 29.5 | 7.7 | 79.1 | 22.5 |
| **Average SR** | **72.5%** | 64.5% | 48.2% | 68.4% | 62.4% |

Table 3: We test PAD performance under various sizes and computational costs.

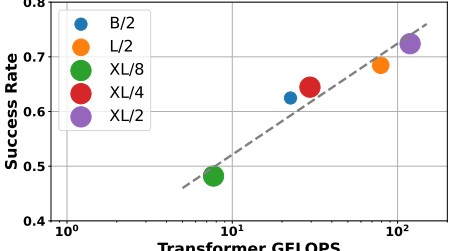

Figure 10: Correlation between Transformer Gflops and policy success rate.

low-level motor control signals [30, 33, 42, 43, 44]. Many studies utilize the reasoning capabilities of pre-trained LLMs and VLMs to create high-level plans followed by motion primitives. Additionally, some approaches adapt pre-trained models to emit low-level motor control signals by adding an action head. Beyond directly acting as policy, pre-trained models can also guide policy learning from multiple aspects, such as providing good representations [45, 46, 47], providing reward signals [48, 49, 50], synthesizing goal images [18, 51], and predicting future sequences [17].

**Diffusion Models for Embodied Control.** Recently, diffusion models have been adopted to tackle challenges in embodied control. A subset of research focuses on training conditional diffusion models that guide behavior synthesis based on desired rewards, and constraints under low dimensional state-input setting [52, 53, 54, 55]. Diffusion Policy [16] trains a visual-motor policy to be conditioned on RGB observations and can better express the multimodal action distributions. However, these methods develop task-specific policies from scratch, missing out on the benefits of pre-training with internet data. Another strand of research utilizes large-scale pre-trained diffusion models to perform data augmentation on training data, such as GenAug [56], ROSIE [57], and CACTI [58].

**Future Prediction for Policy Learning.** There also exist works that leverage future image predictions to assist policy learning. GR-1 [36] employs an autoregressive transformer to sequentially predict future images and actions. In contrast, we adopt a joint diffusion architecture that predicts more accurate future images, potentially leading to improved policy learning performance. UniPi[17] and SuSIE [18] employ a two-stage policy learning process, initially using a diffusion generative model to forecast future image or video sequences, and subsequently training an inverse dynamics model or a low-level diffusion policy based on these goal images. In contrast to these two-stage methods, our approach presents distinct advantages. First, while previous methods utilize diffusion models with a CNN-based U-net backbone [23], designed primarily for image generation and limited to visual predictions, our method adopts a diffusion transformer (DiT) architecture [20]. This architecture adeptly handles multiple modalities concurrently via straightforward token concatenation and attention-mask mechanisms, enabling us to jointly predict future and actions simultaneously. Secondly, using images as the interface between prediction and action may not fully leverage the encoded features inside pre-trained diffusion models. The effectiveness of these two-stage methods depends heavily on the quality of the generated images. In contrast, our model integrates image generation and robotic action within a unified denoising process.

# 6 Conclusion and Discussion

We present PAD, a novel framework to predict future images and generate actions under a joint denoising process. Moreover, PAD can co-train with internet video datasets and extend to other robotic modalities. Both simulated and real-world experiments demonstrated the efficiency of PAD.

A limitation of the current method is that we only tested with three types of modalities. Subsequent endeavors could extend this framework to incorporate additional robot-related input data, such as tactile information, which we believe are valuable research directions. Another limitation is that the control frequency of PAD is not very high since we need to jointly denoise the images and actions. Future work can explore efficient ways to leverage image predictions, such as utilizing the intermediate latent space of predicted images rather than the high-dimensional pixel spaces.

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

# A Appendix

Videos of PAD can be found at `https://sites.google.com/view/pad-paper`.

## A.1 Additional Implementation Details of PAD

### A.1.1 Input Encoder and Output Decoders

**Image Encoder and Tokenizer.** The image encoder is a frozen VAE encoder same as [20]. Take PAD-XL/2 model for example, the encoded latent space for $256 \times 256$ image is $32 \times 32 \times 4$, and patchify into $(32/2) * (32/2) = 256$ patches, which then are tokenized into 256 tokens.

**Robot action Encoder and Tokenizer.** The robot action is concatenated into a vector and passed into MLP layers, we predict $k$ steps of the future each with 7-dimensional poses, which totally consists $(k + 1) * 7$ dimensional vectors ($(k + 1) * 4$ in Metaworld), and then this vector is tokenized into 1 token.

**Depth image Encoder and Tokenizer (If presented).** We directly down-sample the depth image to a size of $32 * 32 * 1$, and follow the same patchfy process as the RGB image. The patch size is set to 8. The patchfy for depth image resulted in $(32/8) * (32/8) = 16$ patches, which then are tokenized into 16 tokens.

**Output Decoder.** The decoder part mainly inverses the encoder part. The decoder process first reconstructs the future latent from the token output by DiT, then adopts the corresponding decoder to recover the original samples in each modality.

## A.2 Additional Implementation Details of Baselines

The RT-1 baseline is based on official implementation `https://github.com/google-research/robotics_transformer`.

The Diffusion Policy baseline is based on `https://github.com/real-stanford/diffusion_policy`, and we follow `https://github.com/real-stanford/scalingup` to add language condition.

The RT-2 baseline is re-implemented by ourselves. We use the InstructBlip-vicuna-7b model as backbone `https://huggingface.co/Salesforce/instructblip-vicuna-7b`.

The GR-1 baseline is built on `https://github.com/bytedance/GR-1`. Since we can not access the pretraining dataset in the original paper, we initialize the model with the author's open-source checkpoint.

### A.2.1 Additional Model Training Details

|  | PAD-XL/2 | PAD-XL/4 | PAD-XL/8 | PAD-L/2 | PAD-B/2 |
|---|---|---|---|---|---|
| **Layers** | 28 | 28 | 28 | 24 | 12 |
| **Hidden size** | 1152 | 1152 | 1152 | 1024 | 768 |
| **Heads** | 16 | 16 | 16 | 16 | 12 |
| **Parameters** | 661M | 661M | 661M | 449M | 128M |
| **Gflops** | 119.1 | 29.5 | 7.7 | 79.1 | 22.5 |
| **Learning Rate** | 1e-4 | 1e-4 | 1e-4 | 1e-4 | 1e-4 |
| **Batch size** | 256 | 256 | 256 | 256 | 256 |
| **Input image shape** | 256*256 | 256*256 | 256*256 | 256*256 | 256*256 |
| **Input noised latent** | 32*32*(4*4) | 32*32*(4*4) | 32*32*(4*4) | 32*32*(4*4) | 32*32*(4*4) |
| **Patchify size** | 2*2 | 4*4 | 8*8 | 2*2 | 2*2 |
| **Image token size** | 256 | 64 | 16 | 256 | 256 |
| **Input robot action shape** | 1*28 | 1*28 | 1*28 | 1*28 | 1*28 |
| **Action token size** | 1 | 1 | 1 | 1 | 1 |
| **Total token size** | 257 | 65 | 17 | 257 | 257 |

Table 4: Models with various size and computational cost.

### A.3 Real world Experiment Details

**Expert data collection.** We collected data on 6 categories of tasks, including button press, cable-route, pick and place, and drawer open/close. For the pick task, we include 4 colors of blocks. For the place task, we include 3 colors of plate. We randomly placed 1-5 objects on the table and asked the robot to pick/place certain objects conditioned on instruction. Some samples of expert demonstrations are visualized in Figure 11.

**Generalization Test Task Samples.** We test the generalization ability of learned policy under numerous unseen objects. The unseen task is much more complicated than expert tasks, as shown in Figure 12. For convenience, videos of PAD can be found at `https://sites.google.com/view/pad-paper`.

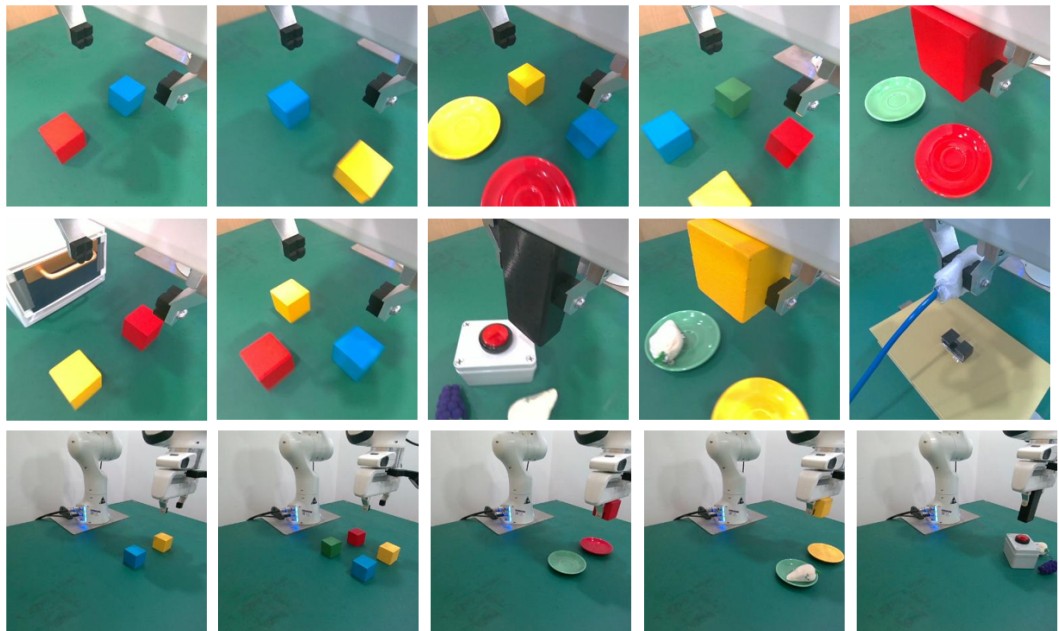

Figure 11: Samples of tasks that we collected demonstrations.

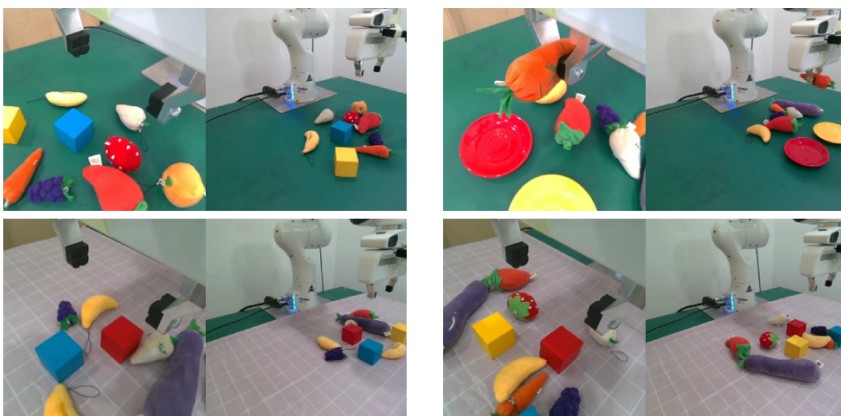

Figure 12: Samples of **unseen** tasks used for generalization test.

## A.4 Details Baselines and Ablations in Metaworld

| Task | Diffusion Policy | SuSIE | RT-1 | RT-2* | GR-1 | PAD | PAD w/o img | PAD w/o cotrain |
|------|------------------|-------|------|-------|------|-----|-------------|-----------------|
| assembly-v2 | 0.20 | 0.40 | 0.00 | 0.24 | 0.64 | **0.88** | 0.04 | 0.32 |
| basketball-v2 | 0.08 | 0.24 | 0.00 | 0.08 | 0.08 | **0.84** | 0.44 | 0.28 |
| button-press-topdown-v2 | 0.16 | 0.32 | **1.00** | 0.84 | **1.00** | 0.92 | 0.92 | 0.92 |
| button-press-top-wall-v2 | 0.00 | 0.24 | 0.60 | 0.88 | 0.80 | **0.96** | 0.64 | 0.84 |
| button-press-v2 | 0.92 | 0.96 | 0.88 | **1.00** | **1.00** | **1.00** | **1.00** | **1.00** |
| button-press-wall-v2 | 0.44 | 0.60 | **1.00** | 0.44 | 0.48 | 0.68 | 0.48 | 0.60 |
| coffee-button-v2 | 0.64 | 0.80 | **1.00** | **1.00** | **1.00** | **1.00** | **1.00** | **1.00** |
| coffee-pull-v2 | 0.00 | 0.32 | 0.08 | 0.68 | 0.52 | **0.80** | 0.40 | 0.32 |
| coffee-push-v2 | 0.16 | 0.32 | 0.04 | **0.76** | 0.44 | 0.52 | 0.48 | 0.64 |
| dial-turn-v2 | 0.52 | 0.52 | 0.00 | 0.52 | 0.44 | **0.56** | 0.28 | 0.52 |
| disassemble-v2 | 0.00 | 0.72 | 0.00 | 0.12 | 0.40 | **0.88** | 0.24 | 0.64 |
| door-close-v2 | 0.52 | 0.56 | **1.00** | **1.00** | **1.00** | **1.00** | **1.00** | 0.96 |
| door-open-v2 | 0.32 | 0.68 | 0.56 | 0.96 | **1.00** | **1.00** | 0.88 | 0.92 |
| drawer-close-v2 | 0.52 | 0.88 | **1.00** | 0.96 | 0.96 | **1.00** | 0.88 | 0.96 |
| drawer-open-v2 | 0.36 | 0.60 | 0.56 | 0.92 | **1.00** | **1.00** | **1.00** | **1.00** |
| faucet-open-v2 | 0.64 | 0.36 | 0.88 | **1.00** | **1.00** | **1.00** | 0.40 | **1.00** |
| faucet-close-v2 | 0.76 | 0.56 | **1.00** | 0.96 | 0.96 | **0.92** | 0.92 | 0.92 |
| hammer-v2 | 0.08 | 0.04 | 0.00 | 0.20 | 0.48 | **0.80** | 0.48 | 0.72 |
| handle-press-side-v2 | 0.44 | 0.72 | **1.00** | 0.76 | 0.48 | 0.40 | 0.76 | 0.40 |
| handle-press-v2 | 0.72 | 0.68 | **1.00** | 0.96 | 0.80 | 0.80 | 0.96 | 0.80 |
| lever-pull-v2 | 0.20 | 0.32 | 0.32 | 0.20 | **0.4** | 0.32 | 0.28 | 0.08 |
| peg-insert-side-v2 | 0.16 | 0.24 | 0.00 | 0.12 | 0.24 | 0.68 | 0.16 | **0.92** |
| peg-unplug-side-v2 | 0.28 | 0.20 | 0.32 | 0.48 | 0.32 | **0.60** | 0.16 | 0.20 |
| pick-out-of-hole-v2 | 0.00 | 0.16 | 0.00 | 0.24 | 0.24 | **0.52** | 0.00 | 0.36 |
| pick-place-wall-v2 | 0.36 | 0.24 | 0.00 | 0.32 | 0.48 | **0.92** | 0.36 | 0.68 |
| pick-place-v2 | 0.08 | 0.24 | 0.00 | 0.48 | 0.72 | **0.80** | 0.20 | 0.20 |
| plate-slide-v2 | 0.60 | 0.68 | 0.08 | **0.88** | 0.88 | 0.72 | 0.16 | 0.48 |
| plate-slide-side-v2 | **1.00** | **1.00** | 0.84 | **1.00** | **1.00** | 0.88 | 0.36 | 0.68 |
| plate-slide-back-v2 | 0.00 | 0.80 | 0.20 | 0.72 | 0.36 | **1.00** | 0.36 | 0.04 |
| plate-slide-back-side-v2 | 0.00 | **0.16** | 0.00 | 0.12 | 0.00 | 0.12 | 0.00 | 0.12 |
| soccer-v2 | 0.00 | 0.08 | 0.28 | 0.28 | 0.12 | 0.08 | 0.08 | **0.32** |
| stick-push-v2 | 0.12 | 0.20 | 0.00 | 0.12 | 0.60 | **0.96** | 0.24 | 0.88 |
| stick-pull-v2 | 0.00 | 0.16 | 0.00 | 0.00 | 0.44 | **0.88** | 0.16 | 0.40 |
| push-wall-v2 | 0.08 | 0.16 | 0.00 | 0.68 | 0.32 | **0.84** | 0.20 | 0.44 |
| push-v2 | 0.20 | 0.32 | 0.00 | 0.40 | 0.44 | 0.52 | 0.16 | **0.60** |
| reach-wall-v2 | 0.72 | 0.92 | 0.12 | 0.76 | **1.00** | **1.00** | 0.92 | 0.92 |
| reach-v2 | 0.40 | 0.60 | 0.76 | 0.40 | **1.00** | **1.00** | 0.44 | **1.00** |
| shelf-place-v2 | 0.00 | 0.08 | 0.00 | 0.20 | 0.28 | **0.72** | 0.16 | 0.56 |
| sweep-into-v2 | 0.08 | 0.16 | 0.12 | 0.36 | 0.16 | **0.40** | 0.24 | 0.28 |
| sweep-v2 | 0.04 | 0.16 | 0.24 | 0.60 | 0.72 | **0.80** | 0.20 | 0.44 |
| window-open-v2 | 0.60 | 0.68 | **1.00** | **1.00** | **1.00** | 0.92 | **1.00** | 0.96 |
| window-close-v2 | 0.36 | 0.96 | **1.00** | 0.96 | **1.00** | **1.00** | **1.00** | 0.96 |
| bin-picking-v2 | 0.00 | 0.24 | 0.00 | 0.12 | 0.40 | **0.88** | 0.00 | 0.72 |
| box-close-v2 | 0.00 | 0.20 | 0.00 | 0.00 | 0.40 | **0.52** | 0.16 | 0.60 |
| door-lock-v2 | 0.12 | 0.32 | 0.00 | 0.40 | 0.60 | **0.88** | 0.12 | 0.72 |
| door-unlock-v2 | 0.44 | 0.44 | 0.28 | 0.52 | 0.44 | **0.60** | 0.68 | 0.56 |
| hand-insert-v2 | 0.08 | 0.24 | 0.16 | 0.28 | 0.20 | **0.40** | 0.08 | 0.20 |
| push-back-v2 | 0.00 | 0.04 | 0.00 | 0.20 | **0.52** | 0.28 | 0.00 | 0.52 |
| Average | 27.9% | 41.0% | 34.6% | 52.2% | 57.4% | **72.5%** | 43.6% | 59.2% |

Table 5: Detailed success rate of baselines and ablations. We did not include the handle-pull-side-v2 and handle-pull-v2 tasks since the expert policy for these two tasks in the original benchmark had low success rates. Every task is tested with 25 rollouts.

## A.5   Detailed Scaling Results

| Task | XL/8 | B/2 | L/2 | XL/4 | XL/2 |
|---|---|---|---|---|---|
| assembly-v2 | 0.32 | **0.96** | **0.96** | 0.20 | 0.88 |
| basketball-v2 | 0.00 | 0.36 | 0.32 | 0.40 | **0.84** |
| button-press-topdown-v2 | **1.00** | 0.76 | **1.00** | **1.00** | 0.92 |
| button-press-topdown-wall-v2 | 0.72 | **1.00** | 0.90 | 0.90 | 0.96 |
| button-press-v2 | **1.00** | **1.00** | **1.00** | **1.00** | **1.00** |
| button-press-wall-v2 | **0.88** | 0.56 | 0.60 | 0.40 | 0.68 |
| coffee-button-v2 | **1.00** | **1.00** | **1.00** | **1.00** | **1.00** |
| coffee-pull-v2 | 0.32 | 0.10 | 0.30 | 0.68 | **0.80** |
| coffee-push-v2 | 0.28 | 0.60 | 0.40 | 0.52 | **0.52** |
| dial-turn-v2 | 0.28 | 0.40 | **0.56** | 0.20 | 0.56 |
| disassemble-v2 | 0.52 | 0.80 | 0.92 | **1.00** | 0.88 |
| door-close-v2 | **1.00** | **1.00** | **1.00** | **1.00** | **1.00** |
| door-open-v2 | 0.92 | **1.00** | **1.00** | **1.00** | **1.00** |
| drawer-close-v2 | 0.52 | **1.00** | **1.00** | **1.00** | **1.00** |
| drawer-open-v2 | 0.88 | **1.00** | **1.00** | **1.00** | **1.00** |
| faucet-open-v2 | 0.32 | **1.00** | **1.00** | **1.00** | **1.00** |
| faucet-close-v2 | 0.92 | 0.44 | 0.96 | **1.00** | 0.92 |
| hammer-v2 | 0.60 | **1.00** | 0.68 | 0.92 | 0.80 |
| handle-press-side-v2 | 0.40 | 0.48 | **0.72** | 0.60 | 0.40 |
| handle-press-v2 | 0.48 | 0.92 | 0.84 | 0.80 | **0.96** |
| lever-pull-v2 | 0.00 | 0.20 | 0.64 | 0.12 | **0.32** |
| peg-insert-side-v2 | **0.80** | 0.52 | **0.80** | **0.80** | 0.68 |
| peg-unplug-side-v2 | 0.44 | 0.32 | 0.56 | 0.48 | **0.60** |
| pick-out-of-hole-v2 | 0.36 | 0.10 | 0.16 | 0.50 | **0.52** |
| pick-place-wall-v2 | 0.20 | 0.68 | 0.56 | 0.60 | **0.92** |
| pick-place-v2 | 0.32 | 0.32 | 0.76 | 0.52 | **0.80** |
| plate-slide-v2 | 0.52 | **0.76** | 0.60 | 0.48 | 0.72 |
| plate-slide-side-v2 | 0.40 | **0.96** | 0.52 | 0.92 | 0.88 |
| plate-slide-back-v2 | 0.20 | 0.36 | 0.12 | 0.20 | **1.00** |
| plate-slide-back-side-v2 | 0.04 | **0.20** | 0.04 | 0.12 | 0.12 |
| soccer-v2 | 0.12 | 0.00 | 0.32 | 0.24 | **0.08** |
| stick-push-v2 | **1.00** | **1.00** | **1.00** | **1.00** | 0.96 |
| stick-pull-v2 | 0.32 | **0.92** | 0.64 | 0.40 | 0.88 |
| push-wall-v2 | 0.60 | 0.40 | **0.84** | 0.80 | **0.84** |
| push-v2 | 0.28 | 0.32 | 0.92 | 0.64 | **0.52** |
| reach-wall-v2 | **1.00** | **1.00** | **1.00** | **1.00** | **1.00** |
| reach-v2 | **1.00** | **1.00** | **1.00** | **1.00** | **1.00** |
| shelf-place-v2 | 0.40 | 0.72 | **0.90** | 0.80 | 0.72 |
| sweep-into-v2 | 0.12 | 0.40 | 0.48 | 0.36 | **0.40** |
| sweep-v2 | 0.48 | 0.64 | **0.88** | 0.80 | 0.80 |
| window-open-v2 | 0.32 | **1.00** | **1.00** | 0.90 | 0.92 |
| window-close-v2 | 0.28 | **1.00** | **1.00** | 0.72 | **1.00** |
| bin-picking-v2 | 0.88 | 0.72 | 0.80 | **1.00** | 0.88 |
| box-close-v2 | 0.40 | 0.40 | **0.72** | 0.52 | 0.52 |
| door-lock-v2 | 0.60 | **0.88** | 0.76 | 0.84 | **0.88** |
| door-unlock-v2 | 0.40 | **0.68** | 0.30 | 0.64 | 0.60 |
| hand-insert-v2 | 0.12 | 0.00 | 0.16 | 0.20 | **0.40** |
| push-back-v2 | 0.16 | 0.20 | **0.52** | 0.12 | 0.28 |
| handle-pull-side-v2 | N/A | N/A | N/A | N/A | N/A |
| handle-pull-v2 | N/A | N/A | N/A | N/A | N/A |
| Average | 48.2% | 62.4% | 68.4% | 64.5% | **72.5%** |

Table 6: Detailed success rate under different model sizes and computational allocations. We did not include the handle-pull tasks since the expert policy for these two tasks are in low success rates.

## A.6 Instructions used in tasks

| Task | Instructions |
|------|-------------|
| assembly-v2 | assemble the object |
| basketball-v2 | shoot the basketball |
| button-press-topdown-v2 | press the button |
| button-press-topdown-wall-v2 | press the button |
| button-press-v2 | press the button |
| button-press-wall-v2 | press the button |
| coffee-button-v2 | press the button |
| coffee-pull-v2 | pull back cup |
| coffee-push-v2 | push forward cup |
| dial-turn-v2 | turn the dial |
| disassemble-v2 | disassemble the object |
| door-close-v2 | close the door |
| door-open-v2 | open the door |
| drawer-close-v2 | close the drawer |
| drawer-open-v2 | open the drawer |
| faucet-open-v2 | open the faucet |
| faucet-close-v2 | close the faucet |
| hammer-v2 | pick up hammer |
| handle-press-side-v2 | press the handle |
| handle-press-v2 | press the handle |
| lever-pull-v2 | pull the lever |
| peg-insert-side-v2 | insert the peg |
| peg-unplug-side-v2 | unplug the peg |
| pick-out-of-hole-v2 | pick red object |
| pick-place-wall-v2 | pick red object |
| pick-place-v2 | pick red object |
| plate-slide-v2 | slide forward plate |
| plate-slide-side-v2 | slide side plate |
| plate-slide-back-v2 | slide back plate |
| plate-slide-back-side-v2 | slide back plate |
| soccer-v2 | kick the soccer |
| stick-push-v2 | push the stick |
| stick-pull-v2 | pull the stick |
| push-wall-v2 | push the object |
| push-v2 | pick red object |
| reach-wall-v2 | reach red object |
| reach-v2 | reach red object |
| shelf-place-v2 | place blue object |
| sweep-into-v2 | sweep brown box |
| sweep-v2 | sweep brown box |
| window-open-v2 | open the window |
| window-close-v2 | close the window |
| bin-picking-v2 | pick green object |
| box-close-v2 | close the box |
| door-lock-v2 | lock the door |
| door-unlock-v2 | unlock the door |
| hand-insert-v2 | put box into hole |
| handle-pull-side-v2 | pull the handle |
| handle-pull-v2 | pull the handle |
| push-back-v2 | push back object |

Table 7: Instructions for each tasks in metaworld. We mainly designed the instruction based on the task names.

