# OpenReview forum: "Prediction with Action: Visual Policy Learning via Joint Denoising Process"
_NeurIPS.cc/2024/Conference — NeurIPS 2024 poster_

### Official Review · Reviewer_GDHG · 2024-06-29

**Soundness:** 2
**Presentation:** 3
**Contribution:** 2
**Rating:** 4
**Confidence:** 5

**Summary:**

The submission aims to enhance policy learning through co-training with image prediction. The intuition is that image prediction and robot action are highly correlated since they share the same underlying physical dynamics. Thus, a diffusion model that generates both future images and actions may generalize better to the task distribution.

As a solution, the paper proposes a diffusion transformer-based solution that takes as input a noisy sequence of future actions and observations and predicts the corresponding noise, in order to return a clean action trajectory and observation sequence. This enables co-training with actionless videos, in which case the action denoising loss is deactivated.

The results show improved language-conditioned multi-task learning on MetaWorld, as well as generalization to different real-world scenarios. Notably, in the real world, the proposed policy is able to generalize to more difficult task variations than the ones seen during training.

**Strengths:**

1. (Presentation) The paper is very clearly written and is very easy to follow and understand. I want to additionally endorse the tone of this paper, which is humble and moderate, without overpresenting or overlaiming.

2. (Contribution) As a technical contribution, the paper shares interesting architectural details on how to repurpose DiT for joint action and future image prediction.

3. (Soundness) The ablations show the importance of co-training with image prediction and the additional benefit of training on video corpora.

I believe that overall the paper contributes to the idea that predicting the future and acting is a promising path for generalizable robot learning, as opposed to action-centric policy learning. Although the idea is not original per se, the paper provides more evidence for it.

**Weaknesses:**

1. (Contribution) The motivation of the paper is that first, the ability to predict future images is coupled with the ability to predict actions (idea); second, diffusion models have shown good results on both tasks, so it is reasonable to combine them for joint prediction (implementation). To the best of my knowledge, the implementation part is novel, but the idea part is not.
Specifically, recent works such as [GR-1] have also argued and supported that jointly predicting future frames and actions is helpful, however there is no discussion and comparison against such an approach.

2. (Soundness) A major concern about this submission is the inadequate evaluation protocol. Specifically, the benchmark used is very simple and important baselines are missing. In more detail:

* The baselines considered in this paper are not carefully selected and important strong baselines are missing. On the first part, it’s not surprising that 2D Diffusion Policy is not weak, but probably the point is to show that using DiT is an improvement over it. RT-1 is also a rather weak baseline without the large data support, as shown in [GR-1]. Moreover, comparing against RT-1 and RT-2 has little to offer as their training paradigm is simply behavior cloning with regression. It would be more useful to compare against other diffusion policies or video prediction models, since the paper's main argument does not stem from its architectural improvements on Transformer-based policies.
On the second part, the main argument of the paper is that predicting both actions and future images helps control. A similar argument is also presented in [GR-1], although their architecture is not diffusion-based but autoregressive. It could be that diffusion (or the proposed architecture in general) is indeed a better solution, but to conclude that, a direct comparison with [GR-1] should be conducted.

* While the end goal is to have robots operating in the real world, it is unsafe to compare and draw conclusions about models’ performance with real-world experiments only. Simulation benchmarks offer some proxy to evaluate performance and compare methods in identical setups. The paper does a good job of presenting both simulation and real-world results. While the community hasn’t concluded on a robotics benchmark that covers many cases and tasks, MetaWorld is definitely one of the simplest benchmarks, without distractors, visual variations or the necessity of modeling geometry. Multi-tasking results on MetaWorld are not very impressive since the task can be usually inferred by just looking at the scene. On that aspect, training on videos from the web or external data (such as BridgeData-v2) to test on MetaWorld feels rather disconnected. It would be more valuable to show results on benchmarks like CALVIN, LIBERO or RLBench, that test more generalization factors and offer more challenging setups with stronger baselines. Specifically, SuSIE, which is a useful baseline as it also predicts future images, as well as [GR-1], which should be the main baseline, both report results on CALVIN. Therefore, evaluating on CALVIN would give direct answers on these comparisons.


3. (Presentation) Although the clarity of presentation is remarkable, the paper lacks a clear take-home message. If this is that hoint action-frame prediction is helpful, then it reduces novelty, since there is already evidence on that. Another suggestion is the following. There are some works (such as the cited Diffuser [48]) that jointly denoise states and actions. This is a relevant background for this work, which denoises observations and actions. The proposed approach is more general since states cannot be accessed in the real world. It would be nice to see some discussion on the connection between prior work and this work, as it would provide additional motivation.

[GR-1] Unleashing Large-Scale Video Generative Pre-training for Visual Robot Manipulation, ICLR24

**Questions:**

Overall, while this submission has merits, its poor evaluation setup, as well as the absence of comparisons and discussions against important prior work makes me reluctant towards acceptance. I would increase my score if:

* The writing changes to include discussion on previous works that had the same high-level idea and focus on the differences with those.

* The evaluation becomes stronger by adding baselines and testing on a more challenging benchmark. On the latter, one suggestion is to evaluate on CALVIN, so as to enable direct comparison to [GR-1] and SuSIE.


--------------
Post-rebuttal: score updated from 2 to 4.

**Limitations:**

Addressed

---

> ### Author Rebuttal · Authors · 2024-08-07
>
> We sincerely appreciate your time and efforts in reviewing our paper! Based on your review, we added a detailed discussion and additional experiments. **Updated Figures can be found in the PDF attached to the global response.**
>
> ---
>
> **Q1: About the contribution and novelty:  the implementation part is novel, but the idea part is not. Specifically, recent works such as [GR-1] have also argued and supported that jointly predicting future frames and actions is helpful.**
>
> ANS: We fully agree that integrating video prediction with action, and more broadly, learning enhanced representations from videos to improve policy learning, is a well-established area of research. In this paper, we focus on **exploring joint multi-modal predictions and actions within a unified diffusion architecture.** This approach, which we all recognize as novel, differentiates our work from existing methods.
> GR1 utilizes an autoregressive model for predicting images and actions. However, recent advancements have shown that **diffusion models outperform autoregressive methods in generating images and videos** [1,2]. Despite these advancements, performing joint denoising processes using the typical U-Net architecture, as seen in prior studies like SUSIE and uniPi, poses significant challenges. Our approach overcomes these challenges by integrating predictions and actions into advanced Diffusion Transformers (DiT), which effectively encode physical knowledge from diverse datasets. **This solution has been recognized as both elegant and nice by all three other reviewers.** A more detailed experimental comparison with GR1 can be found in the subsequent Q&A sections.
>
> ---
>
> **Q2: About the meta world benchmark and Calvin benchmarks**
>
> ANS: We would like to say that numerous previous works [] perform experiments on Metaworld and these two benchmarks present their own unique challenges. Metaworld offers a much more diverse array of tasks requiring precise manipulation and the ability to handle multiple tasks simultaneously. In contrast, Calvin focuses primarily on instruction-following skills, involving fewer objects (just three colors of blocks, one door, one drawer, and one light).
>
> We observed that previous methodologies have already achieved success rates nearing or exceeding 90% on Calvin benchmarks, with most failures occurring in corner cases. Given the limited time available for rebuttal, it would be exceedingly difficult for us to establish a new environment and adjust hyperparameters to surpass this 90% success rate. Furthermore, Calvin shares many similarities with **our real-world experiments which involve even more complex generalization tasks** including brand-new objects and backgrounds. Considering these factors and the tight timeline, we opted to compare the official GR1 implementation with PAD on the Metaworld and the real world. (The real-world experiment is running since it takes time to deploy and adjust real systems.)
>
> **Q3: Detailed experimental comparison with GR-1 baseline**
>
> ANS: Following the GR-1 open-sourced code, we initialize the GR-1 model with their pre-trained checkpoint and perform data-efficient imitation learning on the multitask Metaworld. The comparisons are shown below. Firstly, we observed that the **image prediction quality of PAD significantly surpasses that of GR-1**, likely due to the superior capabilities of diffusion models in image generation tasks. As depicted in the global PDF, the visual prediction results from GR-1 appear blurry, while PAD produces detailed and precise imagery. (Notably, the images presented in the original GR-1 paper were also blurry.) We hypothesize that the **high-quality images generated by PAD can more effectively facilitate policy learning**, leading to higher success rates, particularly in tasks requiring precise and accurate operation such as picking small blocks, insertion, basketball, etc., in Metaworld. We hope this detailed analysis provides insight into why our exploration of joint prediction and action under diffusion architecture is both meaningful and valuable.
>
> | MetaWorld-50tasks | GR-1 | PAD  |
> | ----------------- | ---- | ---- |
> | Success Rate      | 54.6 | 72.5 |
>
> ---
>
> **Q4: The baselines considered in this paper are not carefully selected. It’s not surprising Diffusion Policy is weak. RT1RT2 only utilizes simple BC loss**.
>
> ANS: We respectfully present an alternative perspective. We believe these baselines are both recent and robust, as suggested by Reviewer Hcpx. Diffusion policy is a sample-efficient imitation learning algorithm recognized as a significant breakthrough in robotic policy learning. Although RT1 and RT2 are trained only with BC loss, they are built upon large vision-language models that have already been trained on extensive datasets, demonstrating impressive multi-task capabilities. We compare these baselines to verify the sample efficiency and multi-task learning ability of PAD.
>
> ---
>
> **Q5: Training on videos from the web or external data (such as BridgeData-v2) to test on simulated MetaWorld feels disconnected.**
>
> ANS: Training with videos and testing in simulated environments such as Metaworld is widely adopted in prior research. We recognize a substantial gap between real-world videos and simulations, which has led us to validate our methods through real-world tasks, confirming that co-training significantly improves our models' generalization capabilities.
>
> Despite the noted gap between simulations and reality, as detailed in Section 4.4, **internet videos provide valuable priors and features that enhance image prediction quality in simulations**. The high-quality predicted images may significantly improve policy learning, as it becomes easier for the model to identify robot movements from these precise predicted images.
>
> ---
>
> We hope our response above can solve your concerns and show the improved quality of our paper. We are always ready to answer your further questions!

---

> > ### Comment · Area_Chair_WAJd · 2024-08-12
> > **To reviewer GDHG : Please respond to rebuttal**
> >
> > Hi reviewer GDHG,
> >
> > Thank you for your initial review. Please kindly respond to the rebuttal posted by the authors.
> > Does the rebuttal answer your questions/concerns? If not, why?
> >
> > Best,
> > AC

---

> ### Author Response · Authors · 2024-08-12
> **Welcome to see our new experiments on CALVIN benchmark!**
>
> Dear Reviewer GDHG:
>
> After we submit the initial rebuttal message, we continue to conduct experiments in the Calvin benchmark for better comparisons as you suggested. We are pleased to share some promising results. Following the settings in the GR-1 paper, we conducted imitation learning on the whole training datasets and 10% training datasets respectively.  When using the full dataset, our results were comparable to those reported for GR-1, given that GR-1's original success rate was already high (approaching 100%) and we had limited time for parameter tuning. However, **in the reduced 10% data regimen, PAD clearly outperforms GR-1**, indicating that PAD is a more sample-efficient learning algorithm with joint prediction and action under a unified diffusion architecture. The detailed comparison of task completion numbers in a row is shown below: (some baseline data is sourced from GR-1)
>
> | **Whole datasets** | 1     | 2     | 3     | 4     | 5     | Avg. Len.     |
> | ------------------ | ----- | ----- | ----- | ----- | ----- | ------------- |
> | RT-1               | 0.844 | 0.617 | 0.438 | 0.323 | 0.227 | 2.45          |
> | HULC               | 0.889 | 0.733 | 0.587 | 0.475 | 0.383 | 3.06          |
> | MT-R3M             | 0.752 | 0.527 | 0.375 | 0.258 | 0.163 | 2.08          |
> | GR-1               | 0.949 | 0.896 | 0.844 | 0.789 | 0.731 | **4.21**      |
> | PAD(ours)          | 0.965 | 0.904 | 0.842 | 0.768 | 0.707 | **4.19**      |
> | **10% datasets**   | **1** | **2** | **3** | **4** | **5** | **Avg. Len.** |
> | RT-1               | 0.249 | 0.069 | 0.015 | 0.006 | 0.000 | 0.34          |
> | HULC               | 0.668 | 0.295 | 0.103 | 0.032 | 0.013 | 1.11          |
> | MT-R3M             | 0.408 | 0.146 | 0.043 | 0.014 | 0.002 | 0.61          |
> | GR-1               | 0.778 | 0.533 | 0.332 | 0.218 | 0.139 | 2.00          |
> | PAD(ours)          | 0.813 | 0.606 | 0.412 | 0.294 | 0.189 | **2.31**      |
>
> Thank you once again for your time and effort in reviewing our paper. We hope that these additional experiments address your remaining concerns and demonstrate the improved quality of our work. We are always ready to answer your further questions!

---

> > ### Comment · Reviewer_GDHG · 2024-08-13
> > **Thank you for your answers**
> >
> > Thank you for your effort on the rebuttal.
> >
> > Regarding my concerns on contribution: I also recognized the technical contribution of this paper. My major concern was the absence of related alternatives, specifically GR1, from the discussion. Of course working on an existing problem and advancing it is a valid contribution, however it should be placed in the context of related work. The rebuttal discussed GR1 and I believe it placed itself better in the space of relevant literature.
> >
> > Regarding MetaWorld: while arguing on which benchmark is more challenging is of little value, my take on MetaWorld in my original review was that the task can be usually inferred by just looking at the scene, so that makes multi-tasking results less impressive. This is because of the lack of distractions in the scene, mostly the relevant objects per task are there. CALVIN scenes do not offer a wide variety of tasks to be deployed, but the scene is the same and language understanding is necessary for task execution. At the same time, I thought that evaluation on CALVIN would further allow for easier comparisons against stronger related baselines: SuSIE, which is very related, GR1 and RT1's results are already there for free. Since SuSIE was adapted to MetaWorld, I thought it could be as easy to take its data loader and adapt it to support PAD, thus enabling all those comparisons for free. Lastly, CALVIN offers an established train/test setup, where baselines have shown the benefits of using video data (GR1), future prediction (SuSIE), foundation models (RoboFlamingo). In contrary, MetaWorld papers often define their own setup in terms of tasks and demos used and I'm not aware of a standardized and widely adopted protocol there.
> >
> > I'm eventually glad to see that experiments on CALVIN were ran. I agree that the performance on ABCD->D is already quite high. The SOTA on this split is actually much better than GR1, but showing SOTA results on CALVIN was never my expectation for this submission, the comparison to previous works was. It's good to see that PAD is more sample-efficient. It would be more meaningful to also show results on ABC->D, which is more challenging and could better showcase the advantages of using joint video training.
> >
> > Regarding the baselines: I still believe that GR1 is the most related baseline, then also SuSIE. It's ok to include RT-1 and 2 as well as training paradigms or transformer-based architectures. For diffusion policies, if CALVIN was adopted from the beginning it would allow comparison with 3D Diffuser Actor, which is a much stronger baseline. But it's ok that this is not included as long as other video prediction approaches are included in a fair comparison, which is better satisfied after seeing the CALVIN results.
> >
> > Overall, I appreciate the effort for the rebuttal, especially after seeing the CALVIN results. As more and more video prediction methods for robotics show up, the questions that make the difference are what is the right representation and which implementation details are important. This paper bets on pixels and offers a valid technical contribution. I respect that, even if I'm not fully convinced about the strength of the specific implementation. It's better to let multiple voices be heard, so I'm removing my strong objections against this paper, even if I'm not giving it a full acceptance score for now.
> >
> > Score updated from 2->4.

---

> > > ### Author Response · Authors · 2024-08-13
> > > **Thank you for the comprehensive feedback!**
> > >
> > > Dear Reviewer GDHG:
> > >
> > > Thank you for your comprehensive feedback! We are delighted to hear from you.
> > >
> > > First of all, we are very glad that we reach a consensus on the technical contribution of the paper. As you mentioned, there is a growing integration of video prediction methods in robotic learning. We have bet and explored a unified diffusion architecture, as we all recognized, the first of its kind in this context. Thank you for your suggestions which have helped refine and clarify our contributions!
> > >
> > > As for the CALVIN benchmarks, we totally agree with you that CALVIN can better **evaluate the instruction following ability** and easier to compare with previous works, which had encourage us to conducted additional experiments on it. Thank you for your constructive suggestions!
> > >
> > > Concerning the MetaWorld benchmarks, we would like to offer a perspective that this benchmark is particularly effective at **evaluating the dexterity of the learned policy** due to its requirement for precise actions with numerous small objects. Sometimes robot learning policy can still failed even given a single task to accomplish (as you said, without distractions) due to a lack of precision in execution. We believe that **PAD offers unique advantages in enhancing the dexterity** (or precision) of the policy, as it predicts precise future images and actions through joint denoising. The protocol for this benchmark includes an official scripted policy provided by the authors of MetaWorld to collect demonstrations for each task, which is a method commonly used in previous research.
> > >
> > > In summary, while the Calvin benchmarks prioritize language comprehension and do not focus heavily on dexterity and precision, as the tasks involve only large blocks and objects such as drawers or doors, MetaWorld requires precise execution of dexterous tasks without a strong emphasis on following instructions. We believe that **both language comprehension and dexterity** are crucial for developing intelligent robots. We hope this detailed analysis can explain why our experiments on Metaworld is also meaningful, which assess different aspect of the learned policy.
> > >
> > > Thank you once again for your detailed feedback and the improved score. We will add all the new experiments and discussions to the paper to meet the highest standards. As the rebuttal phase draws to a close, we still remain open to addressing any further queries you may have!
> > >
> > > Best regards,
> > > Authors

---

### Official Review · Reviewer_Hcpx · 2024-07-11

**Soundness:** 2
**Presentation:** 3
**Contribution:** 3
**Rating:** 6
**Confidence:** 3

**Summary:**

- This work proposes a new method for language conditional imitation learning for robotics. The core idea is to combine image diffusion and policy diffusion to simultaneously predict future image observations and actions using a latent diffusion transformer and DDPM.
- The diffusion process is conditioned on the recent image observations, the current robot pose, the language instruction, and for real-word experiments also on a depth map. The output of the diffusion process is the prediction of future image observations, depth observations, if available, and actions for K time steps. The diffusion process is repeated after executing a single action in the environment by conditioning on the updated history.
- The input and output modalities have different dimensionalities which is handled by projecting all modalities into a latent space where the latent diffusion process is applied. A VAE is used for projecting the images, an MLP for the actions, a CLIP encoder for the language instructions, and the depth maps are processed in a similar fashion to the image observations. This setup also allows training the model on video data where no actions are available.
- Experiments are conducted on the simulated Meta-World benchmark and in the real-world with a Panda arm. The proposed method is compared against a multi-task variant of diffusion policy, a modified version of SuSIE, RT-1, and a re-implementation of RT-2 built on top of InstructBlip7B.

**Strengths:**

Originality:
- The paper proposes an interesting combination of image diffusion and policy diffusion to leverage transfer between the two tasks. Using latent diffusion with a transformer so that all modalities can be processed in a shared latent space while also being able to train on video data without actions is a nice and elegant idea.

Quality:
- Experiments on Meta-World and with a real-world Panda arm on both seen and unseen tasks are reasonable for validating the proposed method.
- The selected baselines (a multi-task variant of diffusion policy, a modified version of SuSIE, RT-1, and a re-implementation of RT-2 built on top of InstructBlip7B) are recent and strong.
- For the diffusion policy baseline, a pretrained CLIP encoder was used as in related work which also matches language encoder used by the proposed method. Similarly, a diffusion transformer is used for SuSIE for a fairer comparison.
- Table 1 includes two important ablations: PAD without image diffusion and PAD without on videos without actions that validate two of the main hypotheses of this work. There is also an ablation that shows that using depth maps as additional observations improves performance in real-world experiments.

Clarity:
- The paper is generally well-written and clear.
- Various hyperparameters are provided in the appendix to facilitate reproducibility.
- The videos on the paper website provide a good sense of qualitative performance.

Significance:
- Considering the good empirical performance and the elegance of this approach, it seems reasonable that other researchers and practitioners might use or build on this work.

**Weaknesses:**

Originality:
- The paper lacks a reference to and discussion of DBC introduced in “Diffusion Model-Augmented Behavioral Cloning” by Chen et al. 2024 which first appeared on arXiv in February 2023. Similar to this work, Chen et al. 2024 learn a diffusion model over state-action pairs. Instead of using this directly for parameterising the policy, however, the diffusion model is used for computing an auxiliary loss to guide the policy. This therefore avoids the need to run the computationally expensive diffusion model at test time.
- Several statements are not supported by references:
    - Line 163: “Previous studies that utilized pixel input generally developed separate task-specific policies for each of the 50 tasks“.
    - Lines 285-286: “Many studies utilize the reasoning capabilities of pre-trained LLMs and VLMs to create high-level plans followed by motion primitives.
    - Lines 286-288: “Additionally, some approaches adapt pre-trained models to emit low-level motor control signals by adding an action head.”

Quality:
- Considering the similarities of the proposed method to DBC, it would be good to have DBC as an additional baseline to differentiate this work from DBC more clearly.
- The paper lacks a discussion of previous works that benchmark imitation learning methods on Meta-World and why this particular evaluation methodology was chosen. For example, “PRISE: LLM-Style Sequence Compression for Learning Temporal Action Abstractions in Control” also benchmarks imitation learning approaches on Meta-World, reporting 5-shot performance on five previously unseen tasks. Similarly, the method could also be evaluated on “LIBERO: Benchmarking Knowledge Transfer for Lifelong Robot Learning” which provides tele-operated human demonstrations and several baselines.
- The appendix provides a breakdown of performance across individual tasks and ablations for different model sizes, but all experiments are run with only a single random seed and no error bars or similar are reported.
- It is unclear how the hyperparameters were tuned for the proposed method, for the ablations, and for the baselines. Similarly, it would be useful to have a discussion how the architectures of the proposed method and the baselines compare to better understand the fairness of the comparisons. Table 1 indicates that the proposed method without image diffusion already outperforms three of the four baselines, including diffusion policy. It seems like there are other differences between the proposed method and diffusion policy that account for some of the difference in performance.
- The paper lacks a discussion of how this method compares to the baselines in terms of latency. It is also not clear whether the videos at the provided URL run in real-time. Running image diffusion at every time step seems computationally expensive which might have practical implications.

Clarity:
- None beyond what is mentioned above.

Significance:
- The latency of running image diffusion at every time step is a concern. This might be quite slow and limit the adoption of this work.
- It is not quite clear from the NeurIPS paper checklist whether the code and the datasets are going to be publicly released which would increase the impact of this work.

**Questions:**

- Line 166: How were the Meta-World training trajectories collected? How does this compare to existing literature? Are there existing public datasets of demonstration trajectories for Meta-World?
- How were the hyperparameters tuned for the proposed method, for the ablations, and for the baselines?
- How do the architectures of the proposed method and the baselines compare? Why does the proposed method already significantly outperform diffusion policy even without image diffusion?
- Line 219: How were the tasks that are shown in Table 1 selected?
- Line 21: Do the videos on the paper website show the robot at real-time speed or are the videos accelerated?
- How do the proposed methods and the baselines compare in terms of latency?
- Are code and training data going to be released publicly?

UPDATES AFTER REBUTTAL:
- Presentation: 2 -> 3
- Contribution: 2 -> 3
- Score: 4 -> 6

**Limitations:**

As mentioned above, the paper lacks a discussion of latency considerations which might limit the practical usefulness of this approach.

---

> ### Author Rebuttal · Authors · 2024-08-07
>
> We sincerely appreciate your time and efforts in reviewing our paper! Based on your review, we added a detailed discussion and additional experiments.
>
> ---
>
> **Q1: Discussion and comparisons with DBC paper which learn a diffusion model over state-action pairs as auxiliary loss**
>
> ANS: Thank you for your constructive suggestion! DBC adds the state-action distribution diffusion loss to help policy learning while we use future prediction loss to enhance policy. Despite the lack of open-sourced code from the DBC authors, and their focus on low-dimensional single-task settings using smaller neural networks, we implemented their state-action modeling auxiliary loss within our DiT framework, ensuring comparable parameter sizes. (Detailed implementation framework can be found in global pdf.) In general, we have much larger experiment scales in terms of task numbers, task difficulty, input complexity, and model size. Our results demonstrate that PAD outperforms DBC with a clear margin. We hypothesize this is due to that future prediction loss can better guide policy learning compared to the DBC's state-action distribution loss.
>
> |              | PAD w/o image prediction | DBC            | PAD(ours)      |
> | ------------ | ------------------------ | -------------- | -------------- |
> | Success rate | $$43.4\pm1.7$$           | $$47.6\pm3.0$$ | $$72.3\pm1.8$$ |
>
> ---
>
> **Q2: Several statements are not supported by references.**
>
> ANS: Thank you for pointing out the missing reference! We cite [1,2,3,6] for Line 163 which describes single-tack learning algorithms, cite [7,8] for 285-286 which mentions the VLM high-level plan, and cite [9] for 286-288 mentions the pre-trained model with action head.
>
>
> ---
>
> **Q3: Discussion of previous works that benchmark imitation learning methods on Meta-World and why this particular evaluation was chosen. e.g., PRISE reported 5-shot performance on 5 unseen tasks.**
>
> ANS:   Numerous works in many categories evaluate imitation algorithms on Metaworld benchmark with limited demonstrations, such as diffusion-based algorithms diffusion policy[1], 3D-DP[2], video representation learning algorithms R3M[3], VC-1[4], and language modeling style policy Embodied-GPT[5], ACT[6], etc. Similar to these works, we try to evaluate PAD's imitation capability under limited robotic data and multi-task settings.
>
> PRISE tests the model's few-shot adaptation capability as you said. We follow the exact same pipeline to test the adaptation ability of our PAD model. The results are shown below which further verify the sample efficiency of our method thanks to joint prediction.
>
> |                                         | PRISE          | PAD            |
> | --------------------------------------- | -------------- | -------------- |
> | 5-shot success rate on 5 specific tasks | $$66.8\pm1.8$$ | $$74.2\pm2.3$$ |
>
>
>
> ---
>
> **Q4: How were the Meta-World training trajectories collected? Are there existing public datasets of demonstration?**
>
> ANS: We utilized the scripted expert policies provided by the official Meta-World to collect 50 demonstrations per task, in line with prior studies [1,2,6,12].
>
> ---
>
> **Q5: How were the tasks that are shown in Table 1 selected?**
>
> ANS: The tasks in Meta-World are categorized into different difficulty levels according to [1]. We selected representative tasks from each level to display due to space constraints, and also the average success rate.
>
> ---
>
> **Q6: About random seeds and error bars of experiments.**
>
> ANS: Limited by our computational resources, and given the breadth of ablation studies conducted (ablations on image prediction/co-training, depth input,  and various model sizes), we were unable to run many seeds at the time of submission. We promise to run three seeds for each experiment and update the results in the paper. Moreover, we observed diffusion training process is extremely stable with low variance, corroborating findings from the diffusion policy paper[1].
>
> ---
>
> **Q7: How the hyperparameters were tuned. The architectural differences between the proposed method and diffusion policy**
>
> ANS: Yes, there do exist architectural differences.  Unlike the official diffusion policy, which utilizes CNN-based image encoders and cross-attention conditions, our PAD is built upon the DiT[10] which pathify images into tokens and uses feature-wise modulation for condition. We have integrated additional modules into DiT to support language and multi-modal inputs/outputs, using the same training hyperparameters as the DiT paper. As discussed in previous works[11], DiT is a highly optimized architecture that may also contribute to our performance gain.
>
> ---
>
> **Q8: Do the videos on the paper website show the robot at real-time speed?**
>
> ANS: The old videos are recorded inside the algorithm pipeline which has some bias. We updated some videos recorded by third-view mobile phones which are truly real-time.
>
> ---
>
> **Q9: The latency of running image diffusion at every time step is a concern. How do the proposed methods and the baselines compare in terms of latency?**
>
> ANS: We total agree with you that diffusion-based policies typically exhibit lower control frequencies due to the multiple denoising steps involved.  In real-world robotic control, our model operates at 1-1.5Hz using 50-75 denoising steps, compared to the RT-2 model's 3Hz and the diffusion policy's 3-4 Hz, which works fine in our tested tasks. To further improve the diffusion speed, we can use recent more advanced diffusion accelerated samplers[13] and execute multiple prediction steps as [1].
>
> ---
>
> **Q10: Are code and training data going to be released publicly?**
>
> ANS: We have included an initial code version in the supplementary materials (though a little bit dirty). We will open-source the refined code, dataset, and checkpoints upon acceptance. We are very glad to make contributions to the community!
>
> ---
>
> Thank you again for your time and efforts!  We are always ready to answer your further questions!

---

> > ### Author Response · Authors · 2024-08-08
> > **References for rebuttal**
> >
> > [1] Chi C, Feng S, Du Y, et al. Diffusion policy: Visuomotor policy learning via action diffusion[J]. RSS2023
> >
> > [2] Ze Y, Zhang G, Zhang K, et al. 3d diffusion policy[J]. RSS2024
> >
> > [3] Nair S, Rajeswaran A, Kumar V, et al. R3m: A universal visual representation for robot manipulation[J]. CoRL2022
> >
> > [4] Majumdar A, Yadav K, Arnaud S, et al. Where are we in the search for an artificial visual cortex for embodied intelligence?[J]. Neurips2023
> >
> > [5] Mu Y, Zhang Q, Hu M, et al. Embodiedgpt: Vision-language pre-training via embodied chain of thought[J]. Neurips2023
> >
> > [6] Zhao T Z, Kumar V, Levine S, et al. Learning fine-grained bimanual manipulation with low-cost hardware[J]. RSS2023
> >
> > [7] Ahn M, Brohan A, Brown N, et al. Do as i can, not as i say: Grounding language in robotic affordances[J]. CoRL 2033
> >
> > [8] Driess D, Xia F, Sajjadi M S M, et al. Palm-e: An embodied multimodal language model[J]. arXiv preprint arXiv:2303.03378, 2023.
> >
> > [9] Chen W, Mees O, Kumar A, et al. Vision-language models provide promptable representations for reinforcement learning[J]. arXiv preprint arXiv:2402.02651, 2024.
> >
> > [10] Peebles W, Xie S. Scalable diffusion models with transformers[C]//ICCV 2023: 4195-4205.
> > [11] Esser P, Kulal S, Blattmann A, et al. Scaling rectified flow transformers for high-resolution image synthesis[C]//ICML 2024.
> > [12] Wang H C, Chen S F, Hsu M H, et al. Diffusion model-augmented behavioral cloning[J]. ICML 2024.
> >
> > [13] Song Y, Dhariwal P, Chen M, et al. Consistency models[J]. arXiv preprint arXiv:2303.01469, 2023.

---

> > ### Comment · Reviewer_Hcpx · 2024-08-12
> >
> > Thank you for the rebuttal.
> >
> > Q1: Did you sweep the lambda hyperparameter for weighting the two loss terms in DBC or did you use the value used in the original paper?
> >
> > Q7: For the baselines, did you tune the hyperparameters on the benchmarks that are being used here or did you use the same hyperparameters as in the original papers? For the ablations, did you re-tune other hyperparameters or are they kept the same?

---

> ### Author Response · Authors · 2024-08-12
> **Response to Reviewer Hcpx's further questions**
>
> Dear Reviewer Hcpx:
>
> We are very delighted to hear from you! Here are our replies:
>
> Q1: Did you sweep the lambda hyperparameter for weighting the two loss terms in DBC or did you use the value used in the original paper?
>
> Ans: We used the value of lambda from the original paper. As shown in Figure 6 of the DBC paper, the authors performed a hyperparameter sweep and found that a lambda value around 1 yields the best performance. Consequently, we adopted lambda = 1 for our experiments.
>
> Q7:  For the baselines, did you tune the hyperparameters on the benchmarks  that are being used here or did you use the same hyperparameters as in the original papers? For the ablations, did you re-tune other  hyperparameters or are they kept the same?
>
> Ans: For diffusion policy, we use their config for simulation envs( https://diffusion-policy.cs.columbia.edu/data/experiments/image/lift_ph/diffusion_policy_transformer/config.yaml). We tried to tune the predicti_lens and transformer layer parameters in their config but find their original configs performs the best. For  RT-1, and Susie, we utilized the parameters from their open-source implementations. As for RT-2, since the code was not publicly available, we carefully implemented it based on the descriptions provided in the paper.
>
> Regarding our ablation study, we maintained the same hyperparameters and only removed the specific components being ablated.
>
> ---
>
> We hope these answers address your questions. Thank you again for your valuable time! We are always ready to answer any of your new questions!
>
> Best regards,
>
> Authors

---

> > ### Comment · Reviewer_Hcpx · 2024-08-12
> >
> > > "Q5: How were the tasks that are shown in Table 1 selected? ANS: The tasks in Meta-World are categorized into different difficulty levels according to [1]."
> >
> > I think this might be the wrong reference as [1] does not evaluate on Meta-World?

---

> ### Author Response · Authors · 2024-08-12
> **About the reference**
>
> Dear Reviewer Hcpx:
>
> Sorry for the wrong reference. The correct reference should be [14]. On page 18 of the experiment details of [14], there is a table provided categorizes tasks into easy, medium, hard, and very hard. In our table, the "easier tasks" row includes some of the easy tasks, while the "harder tasks" row encompasses medium, hard, and very hard tasks. This categorization is also employed in the 3D Diffusion Policy paper [15], specifically in Section IV.A, line 17.
>
> We have check all other citations for accuracy.
>
> Thank you for your time, and we greatly appreciate your detailed feedback!
>
> Best regards,
>
> Authors
>
> [14] Seo, Y., Hafner, D., Liu, H., Liu, F., James, S., Lee, K., & Abbeel, P. Masked world models for visual control. In Conference on Robot Learning (pp. 1332-1344). PMLR.
>
> [15] Ze Y, Zhang G, Zhang K, et al. 3d diffusion policy[J]. arXiv preprint arXiv:2403.03954, 2024. RSS2024

---

> ### Author Response · Authors · 2024-08-13
> **Dear Reviewer Hcpx: Thank you for your time and effort!**
>
> Dear Reviewer Hcpx,
>
> Thank you for your interest in our paper and for your detailed feedback over the past few days!
>
> We are extremely grateful for your patient and thoughtful insights, which have significantly improved our manuscript. We will release the code upon acceptance, enabling you to examine the specifics more thoroughly. As it approach the end of the rebuttal phase, if you find the content of our paper and our responses to your questions satisfactory, we would sincerely appreciate any consideration for a score improvement.
>
> Once again, we deeply appreciate your efforts in reviewing our work and for the insightful questions that have been invaluable in enhancing our research. We still remain open to addressing any further queries you may have!
>
> Best regards,
>
> The Authors

---

### Official Review · Reviewer_yjvj · 2024-07-12

**Soundness:** 3
**Presentation:** 3
**Contribution:** 3
**Rating:** 8
**Confidence:** 4

**Summary:**

This submission presents a new learning framework called PAD, which utilizes a diffusion transformer (DiT) to jointly denoise both future image frames (RGB/Depth) and generate actions together. This joint learning process yields a scalable model that achieves higher success rates compared to various other robotics transformer model baselines. The paper shows co-training with internet scale data is not only possible thanks to DiT but also brings about much higher success rates and shows strong visual generalization.

**Strengths:**

- An interesting joint denoising process is presented to generate future frames and actions. The paper further shows the possibility that PAD can be extended to more modalities if the engineering efforts are taken to do so which is great to see.
- Nice use of the diffusion transformer architecture which enables co-training on large video datasets that do not come with action labels, providing internet-scale knowledge that improves success rate. It would be nice to see how PAD without co-training on internet data performs on the generalization benchmark. I would imagine it won't do as well as PAD with internet data but is worth adding in my opinion (not a huge issue though).
- Strong results on generalization benchmarks and solid ablation studies on co-training show great promise about the proposed method.
- PAD also shows good scaling results where larger models/more training yields higher success rates.

**Weaknesses:**

- How well does PAD predict future frames? Only a few next frames are shown in fig 7 but I am curious to see how far it can predict? In some sense tasks where there is a lot more ambiguity (e.g. objects that fall and bounce, or finding an object hidden behind something) would be helpful to help understand how PAD handles uncertainty. The current set of tested tasks are fairly straight forward in terms of predicting the next frames.
- What are the failure modes in the real world of PAD? Why might it not succeed? This is not an easy problem to answer regardless but if possible it would be nice to address somewhere.
- What is the data efficacy of PAD compared to past methods? How many demonstrations are needed to get the results? Can PAD use less demos (perhaps thanks to future frame predictions?).

**Questions:**

Questions:
- Some discussion around how this relates to world models like Dreamer or TDMPC for robotics might be important. A joint denoising process that predicts future frames and actions resembles a lot of world models. The world model like nature could be the reason why performance is higher.
- While PAD generalizes, it is more of a generalization on the visual data side instead of manipulation right? The unseen objects probably are graspable by the same actions you would take to grab the seen objects. Regardless the visual generalization looks good.
-

**Limitations:**

Limitations are very lightly mentioned at the end around how PAD only uses a few modalities. Some limitations around the data efficacy of PAD might be good to mention, especially given the expensive nature of real world robotics demonstrations (especially if you want high success rates).

---

> ### Author Rebuttal · Authors · 2024-08-07
>
> We sincerely appreciate your time and efforts in reviewing our paper! Based on your review, we added a detailed discussion and additional experiments.
>
> ---
>
> **Q1: How well does PAD predict future frames? how far PAD can predict in some sense tasks where there is a lot more ambiguity (e.g. objects that fall and bounce, or finding an object hidden behind something) would be helpful to help understand how PAD handles uncertainty.**
>
> ANS: Thank you for your insightful comments! For better understanding, we have visualized additional results in the attached PDF file of the global response. With joint training on the internet video dataset, PAD shows impressive image prediction capabilities.
>
> ---
>
> **Q2: What are the failure modes in the real world of PAD? Why might it not succeed? This is not an easy problem to answer regardless but if possible it would be nice to address somewhere.**
>
> ANS: This is an interesting question! We summarize some failure modes based on our hundreds of rollouts as follows: (1) unexpected collisions and deformable irregular objects add difficulty to tasks, (2) using a wrist-mounted camera and randomly placing the objects can sometimes cause objects to go out of the frame during execution, leading to failure, and (3) PAD may occasionally mix up similar objects (e.g., a red apple and a red block).
>
>
> ---
>
> **Q3: What is the data efficacy of PAD compared to past methods? How many demonstrations are needed to get the results? Can PAD use less demos (perhaps thanks to future frame predictions?).**
>
> ANS: As detailed in the experiment setups, similar to [x], we collected 50 demonstrations for each task using Metaworld's official scripted policy. To assess the sample efficiency of the PAD method, we performed ablations with 20 demonstrations to test the efficacy of PAD.
>
> |               | RT2  | PAD(ours) |
> | :------------ | ---- | --------- |
> | 20 demos/task | 0.44 | **0.62**  |
> | 50 demos/task | 0.52 | **0.72**  |
>
> In the low data regime, PAD also surpasses the baseline
>
>
> ---
>
> **Q4: Discussion with world models like Dreamer or TDMPC. A joint denoising process that predicts future frames and actions resembles a world model could be the reason why performance is higher.**
>
> ANS: Thank you for the suggestion! We completely agree that PAD is closely related to world models, as both encode physical knowledge. Dreamer and TDMPC are online model-based RL algorithms that learn world models with online collected data (without large-scale internet datasets). Previous works have also suggested that scaling up the model size under an online RL setting can be unstable. In contrast, we ensemble a world model into a unified DiT model, which is readily scalable and co-trained with internet datasets. In experiments, we observed good scaling performance with the proposed PAD model.
>
>
> ---
>
> **Q5: While PAD generalizes, it is more of a generalization on the visual data side instead of manipulation right? The unseen objects probably are graspable by the same actions you would take to grab the seen objects. Regardless the visual generalization looks good.**
>
> ANS: Yes, you are correct! The generalization is more on the visual side, such as with many distractions, unseen objects, and unseen backgrounds. We acknowledge that the current model cannot perform brand-new skills that are not present in the training data. We will add this to the limitations and future work section. Scaling the robotic datasets may lead to the emergence of new skills, which are interesting research directions.
>
>
> ---
>
> Thank you again for your insightful and constructive comments! We hope our response above can help you better understand our paper and we are always ready to answer your further questions!

---

> > ### Comment · Area_Chair_WAJd · 2024-08-12
> > **To reviewer yjvj : Please respond to rebuttal**
> >
> > Hi reviewer yjvj,
> >
> > Thank you for your initial review. Please kindly respond to the rebuttal posted by the authors.
> > Does the rebuttal answer your questions/concerns? If not, why?
> >
> > Best,
> > AC

---

> ### Comment · Reviewer_yjvj · 2024-08-13
> **Response**
>
> Thanks for the additional visualizations. I don't think MetaWorld is hard enough or has any ambiguous like tasks really so it's hard to tell how PAD handles situations where objects may unpredictably move around or occlusions.
>
> I currently do not have any other concerns and will bump up my score to 8 as I don't really share the concerns as some of the other reviewers upon reviewing their comments. I do agree with other reviewers that this idea is not really novel, it's a mix of established techniques and in my opinion is very much an "expected idea". But given it performs much better than related work already, is well worth at minimum a score of 8. I can see other papers citing this paper because they really use it as a baseline which is impactful enough for me.
>
> I leave my confidence at 4 since I am experienced with robot learning (small/large scale) and simulation, although not experienced with diffusion models specifically.

---

> ### Author Response · Authors · 2024-08-13
> **Thank you for your support!**
>
> Dear Reviewer yjvj,
>
> We sincerely appreciate your support and engagement with our work!
>
> Regarding the visualizations, Metaworld's requirement for precise movements means it cannot adequately demonstrate how PAD handles uncertainty. As we also co-train on bridge datasets, we have included visualizations of our model's predictions on bridge at the top of our website (as noted in the abstract), **which better illustrate how PAD manages uncertainty in long-horizon prediction**. Visualization include the following scenarios: (1) After opening the cabinet door, a blue bowl and banana are imagined inside the cabinet. (2) A carrot, previously obscured, appears in the sink, though the ground truth is a pepper. PAD infers the identity of the object based on its exposed portion.
>
> Regarding the novelty, we acknowledge that the high-level idea of robot learning involving prediction and action—more broadly, deriving better representations from video datasets—is a well-explored area. Our contribution and novelty lie in proposing a unified diffusion-based architecture that jointly predicts futures and actions. This approach is inspired by the impressive recent performance of DiT in image/video generation [1,2]. Our findings suggest that DiT can also handle multi-modal predictions in robotic tasks with appropriate technical design.
>
> Thank you once again for your time and effort in reviewing our work! We are committed to continually refining our research to meet the highest standards.
>
> Best regards,
> The Authors
>
> ---
>
> [1] Esser P, Kulal S, Blattmann A, et al. Scaling rectified flow transformers for high-resolution image synthesis[C]//ICML 2024.
>
> [2] Ma X, Wang Y, Jia G, et al. Latte: Latent diffusion transformer for video generation[J]. arXiv preprint arXiv:2401.03048, 2024.

---

### Official Review · Reviewer_EGNn · 2024-07-15

**Soundness:** 3
**Presentation:** 3
**Contribution:** 3
**Rating:** 7
**Confidence:** 3

**Summary:**

The paper presents a novel framework called PAD. This framework unifies image prediction and robot action within a joint denoising process, leveraging Diffusion Transformers to integrate images and robot states. PAD supports co-training on both robotic demonstrations and large-scale video datasets and can be extended to other robotic modalities like depth images. The framework significantly improves performance on the Metaworld benchmark and real-world robot manipulation tasks, demonstrating superior generalization to unseen tasks.

**Strengths:**

- The motivation of this paper is clear. The paper is easy to follow and the proposed framework makes sense to me.  The idea is simple and straightforward but effective.
- The performance looks good compared to the RT-1/RT-2 baselines.
- The experimental section is comprehensive, verifying performance on both sim and real world tasks. The authors also tested the model with various computation costs.
- Implementation details are clearly described and code is provided, which makes the community easy to reproduce.

**Weaknesses:**

- The performance improvement on real data is somewhat marginal, which makes it questionable about its effectiveness on real data. It would also be great to showcase the ablation studies for real-world tasks to understand whether this is caused by limited model size and pretrained data (compared to RT-2).

**Questions:**

- It would be interesting to ablate the co-training data in detail with different sources to understand its effectiveness.

**Limitations:**

The authors briefly discussed the limitation in the last section, which looks good to me.

---

> ### Author Rebuttal · Authors · 2024-08-07
>
> We sincerely appreciate your time and efforts in reviewing our paper! Based on your review, we added a detailed discussion and additional experiments.
>
> ---
>
> **Q1: The performance improvement on real data is somewhat marginal. Is it caused by limited model size and pre-trained data compared to RT-2?**
>
> ANS: Thank you for the insightful question! We would like to clarify that although PAD marginally outperforms RT-2 in real-world seen tasks,  PAD demonstrates a significant advantage in unseen tasks, outstripping RT-2 with an average success rate improvement of 28%. This suggests that PAD’s superior generalization capabilities stem from its integration of physical knowledge via future prediction loss. Additionally, RT2  is already a very strong baseline with its huge 7B parameters while the largest PAD model contains ~670M parameters.  Although PAD shows great scaling results in our ablations,  we could not afford to train a diffusion model with billions of parameters from scratch. Instead, we had tried a smaller version of RT-2 finetuned on the BLIP2-2.7B model, which does not perform as well as its larger 7B counterparts.
>
> | Real-world Taks Success Rate | RT2-Blip2-2.7B | RT2-InstructBlip-7B | PAD(ours) |
> | ---------------------------- | -------------- | ------------------- | --------- |
> | Seen Task                    | 0.61           | 0.69                | **0.72**  |
> | Unseen task                  | 0.24           | 0.31                | **0.58**  |
>
>
>
> ---
>
> **Q2: It would be interesting to ablate the co-training data in detail with different sources to understand its effectiveness.**
>
> ANS: This is an interesting question! We conducted an ablation study focusing on the scale of the video dataset to assess the impact of co-training with Bridge. Our findings suggest that co-training mainly enhances performance on unseen tasks. A possible explanation is that unseen objects from unseen tasks, such as strawberries, plates, and bananas, are already appeared in the extensive Bridge datasets. Therefore, co-training significantly benefits these unseen tasks.
>
> | Real-world Taks Success Rate | PAD w/o bridge-v2 | PAD with 10% bridge-v2 | PAD(ours) |
> | ---------------------------- | ----------------- | ---------------------- | --------- |
> | Seen Task                    | 0.68              | 0.68                   | **0.72**  |
> | Unseen task                  | 0.40              | 0.48                   | **0.58**  |
>
>
> ---
>
> Thank you again for your time and efforts! We show our deepest appreciation for your support of our work.  We are always ready to answer your further questions!

---

> > ### Comment · Area_Chair_WAJd · 2024-08-12
> > **To reviewer EGNn : Please respond to rebuttal**
> >
> > Hi reviewer EGNn,
> >
> > Thank you for your initial review. Please kindly respond to the rebuttal posted by the authors.
> > Does the rebuttal answer your questions/concerns? If not, why?
> >
> > Best,
> > AC

---

> > > ### Comment · Reviewer_EGNn · 2024-08-13
> > >
> > > Thanks for the authors' response. I have also read other reviewers' comments and decide to keep my score as 7

---

> > > > ### Author Response · Authors · 2024-08-13
> > > > **Thank you for your endorsement of our paper!**
> > > >
> > > > Dear Reviewer EGNn:
> > > >
> > > > We would like to express our profound gratitude for your endorsement of our paper. We will keep polish the paper to meet highest standard. Once again, thank you for your time and effort in reviewing our paper!
> > > >
> > > > Best, Authors

---

### Author Rebuttal · Authors · 2024-08-07

We sincerely appreciate the time and efforts of all reviewers and the AC in evaluating our paper. We are grateful for the insightful and constructive suggestions, which have helped us improve our work. Below, we summarize our contributions and updates. (Updated Figures are in the attached PDF.)

**1. Contributions:**

- **Motivation:** The motivation of this paper is clear and interesting [EGNn,yjvj,Hcpx].
- **Method:** The joint prediction and action framework is nice and elegant [yjvj,Hcpx], with novel implementation[GDHG]. It seems reasonable that other researchers and practitioners might use or build on this work [Hcpx].
- **Experiments:** The experimental section is comprehensive including strong and recent baselines, thorough ablations, and good scaling results[EGNn,yjvj,Hcpx]. Implementation details are clearly described and code is provided, which makes the community easy to reproduce[EGNn].
- **Presentation:** The method is effective and easy to follow [EGNn,yjvj,Hcpx,GDHG]. The tone of this paper is humble and moderate, without over-presenting or overclaiming[GDHG].

**2. Modifications:**

- Following reviewer EGNn's suggestion, we have ablated the RT-2 model size and co-train dataset to better verify the effectiveness of our methods in real-world tasks.
- Following reviewer yjvj's suggestion, we conducted ablation studies with even fewer demos (20 demos per task). PAD continues to outperform the baseline in lower data regimes.
- Following reviewer Hcpx's suggestion, we reimplemented the DBC baseline within the DiT framework (Detailed in attached pdf). The results show that future prediction loss more effectively guides policy learning compared to DBC's state-action modeling loss.
- Following reviewer yjvj's suggestions, we tested PAD's 5-shot adaptation abilities using PRISE's settings. PAD surpasses the baseline in this scenario as well.
- Following reviewer GDHG's suggestions, we compared our approach with the official GR-1 baseline. Despite both employing joint future and action prediction, our diffusion-based PAD generates images of much higher quality(visualized in attached pdf) and result in a better policy success rate.
- We have corrected all writing typos and added missing references, thanks to the detailed advice provided.

Considering the limited rebuttal time and numerous experiments proposed by all reviewers, we have tried our best to conduct as many experiments as possible. We believe our comprehensive experiments in both simulated and real-world tasks adequately support our claims.

Once again, we sincerely appreciate the reviewers' time and feedback. We remain ready to address any further concerns!

---

> ### Author Response · Authors · 2024-08-14
> **Welcome any discussion if you want at the last moment!**
>
> Dear Reviewers and Area Chairs,
>
> As the discussion period draws to a close, with less than two hours remaining, we remain available to engage in any further discussions you may wish to have. We deeply appreciate your insights and feedback and are grateful that all reviewers have acknowledged our contributions and novelty to varying extents. We sincerely thank you for the time and effort you have dedicated to reviewing our paper.
>
> Best regards,
>
> Authors

---

### Decision · Program_Chairs · 2024-09-25

**Decision:**

Accept (poster)

**Comment:**

This paper presents PAD, a framework for visual policy learning that combines image prediction and robot action within a joint denoising process using Diffusion Transformers. The approach demonstrates significant improvements over baselines on the MetaWorld benchmark and real-world robot manipulation tasks, showing better generalization to unseen scenarios.
All reviewers acknowledged the technical contribution and potential impact of the work. The main strengths highlighted were the innovative integration of image prediction and action generation, the ability to co-train on robotic demonstrations and large-scale video datasets, and strong performance on both simulated and real-world tasks. Reviewers also praised the clear presentation and comprehensive experiments.

Initial concerns focused on the novelty of the core idea, the choice of baselines and benchmarks, and the lack of comparison to some relevant prior work. The authors provided detailed responses, including additional experiments on the CALVIN benchmark and comparisons to the GR-1 baseline. These responses addressed many of the reviewers' concerns, with two reviewers explicitly improving their scores post-rebuttal.

The authors' willingness to run new experiments on CALVIN during the rebuttal period was particularly well-received, demonstrating PAD's superior sample efficiency compared to GR-1. While some reservations remain about the strength of the specific implementation and the choice of benchmarks, the overall consensus is that PAD represents a valuable contribution to the field of robot learning.

For the final version, I encourage the authors to incorporate their rebuttal explanations, particularly the additional CALVIN results, comparisons to GR-1, and discussions on the complementary nature of the MetaWorld and CALVIN benchmarks. Expanding the related work section to better contextualize PAD within the landscape of video prediction methods for robotics would also strengthen the paper.